# Tough fiber-reinforced composite ionogels with crack resistance surpassing metals

Xiaolin Lyu [1] ✉, Kun Yu[1], Haoqi Zhang[1], Piaopiao Zhou[2] ✉, Zhihao Shen [3] & Zhigang Zou [1,4] ✉

Ion-conductive materials have received much attention because of their good mechanical and electrical properties. However, their practical applications are still hampered by limited toughness and crack resistance, stemming from the restricted size of energy dissipation zones, which impacts their reliability and durability. Herein, tough fiber-reinforced composite ionogels (FRCIs) with crack resistance are fabricated by incorporating high-performance fibers into elastic ionogels to efficiently dissipate energy. The FRCIs exhibit good tearing toughness, high strength, high elastic modulus, and low bending modulus. The toughness and crack resistance of the FRCI far exceed that of previously reported gel materials, even outperforming metals and alloys. Furthermore, the electrical resistance of FRCI shows high sensitivity to deformation. Moreover, it remains undamaged after undergoing 10,000 bending cycles when fixing the artificial bone, and possesses self-sensing impact resistance, demonstrating great potential in intelligent robots and smart protective equipment.

Ion-conductive materials (ICMs), such as ionogels, have garnered considerable attention due to their flexibility and good ionic conductivity, which render them suitable for utilization in frontier fields such as soft robotics, flexible iontronic sensors, actuators, wearable devices, etc.[1–3]. In practical service, ICMs require good toughness, high strength, and high modulus to withstand heavy loads and resist crack propagation, thereby improving their stability and longevity. However, most ICMs still exhibit inadequate toughness (<10 kJ m$^{-2}$), weak strength (<1 MPa), and low modulus (<0.1 MPa), limiting their use in applications requiring mechanical robustness[4–7].

The reinforcing strategies of ICMs typically involve molecular engineering and multiphase design. Molecular engineering toughens ICMs by introducing sacrificial bonds, such as double networks and supramolecular interactions, into the polymer network[8–12]. However, once the sacrificial network is destroyed, the elastic network makes it difficult to further provide robust mechanical support to the ICM. While supramolecular interactions can enhance the strength and

toughness of materials by dissipating energy, the design of ultra-tough ICMs remains limited[13,14]. Multiphase design aims to enhance the toughness and crack insensitivity of homogeneous networks by incorporating reinforced microdomains through microphase separation[15–19]. Additionally, mechanical training and nanocomposites (like MXene and liquid metal) can also promote additional interactions, thereby augmenting the toughness of ICMs[20–22]. Nevertheless, these approaches often require laborious synthesis procedures and may also compromise the flexibility and bending abilities of ICMs[18,23,24]. More importantly, the toughness and crack resistance of these ICMs still pale in comparison to those of established tough materials, such as metals and alloys[25,26]. This is because both molecular-scale design and microscale assembly intrinsically restrict the size of the energy dissipation zone within ICMs (fractocohesive length <10 mm). Therefore, designing flexible yet tough ICMs remains a challenging task.

Herein, the fabrication of tough fiber-reinforced composite ionogels (FRCIs) is achieved by embedding high-performance fibers

[1]Key Laboratory of Advanced Materials Technologies, College of Materials Science and Engineering, Fuzhou University, Fuzhou, China. [2]Department of Critical Care Medicine, Fujian Medical University Union Hospital, Fuzhou, China. [3]Key Laboratory of Polymer Chemistry and Physics of Ministry of Education, College of Chemistry and Molecular Engineering, Peking University, Beijing, China. [4]Eco-materials and Renewable Energy Research Center, College of Engineering and Applied Sciences, Nanjing Uninversity, Nanjing, China. ✉e-mail: lyuxiaolin@fzu.edu.cn; zhoupiaopiao@fjmu.edu.cn; zgzou@nju.edu.cn

into elastic supramolecular ionogels. The ionogel forms a robust binding force with the negative charges on the fiber surface, resulting in a tight adhesion between the fibers and the ionogel and significantly enhancing the strength of FRCI. Furthermore, the ionogel efficiently dissipates the force exerted on the fibers, preventing stress concentration during fracture. This generates a large energy dissipation zone and high toughness. The toughest aramid fiber-reinforced composite ionogel has high strength (365 MPa), good tearing toughness (4219 kJ m$^{-2}$), high elastic modulus (1.0 GPa), and low bending modulus (1.2 MPa). The toughness and crack resistance of the FRCI show great advantages, whether compared to other gel materials (e.g., ionogels and hydrogels) or compared to metals and alloys. In addition, compared to the current state-of-the-art fiber-reinforced elastomers (2500 kJ m$^{-2}$) or fiber-reinforced polyurethanes (2012 kJ m$^{-2}$)[27,28], FRCI also exhibits a further improvement in tearing toughness (1.7–2.1 times). To our knowledge, such toughness surpasses all previously reported materials. Additionally, the FRCI displays ionic conductivity, fast response time, and good strain sensitivity. It remains crack-free after enduring 10,000 bending cycles, effectively securing artificial bones. When used as an impact-resistant material, it can provide protection while simultaneously sensing the time and intensity of impacts. This demonstrates its significant potential in applications such as intelligent robots, wearable devices, and smart protective equipment.

## Results

### Material design

The FRCIs were produced in a mold using high-performance fibers and ionogels (Supplementary Fig. 1). To elucidate our unique design, the ionogel composed of poly(acrylic-co-acrylamide) (P(AA-co-AAm)) and 1-ethyl-3-methylimidazolium trifluoromethanesulfonate ([EMIM][OTf]) was chosen as the model ionogel. Meanwhile, carbon fiber (CF) was used as the fabric in the FRCI since CF has the advantages of low density, high strength, high modulus, and good temperature tolerance (Supplementary Tables 1 and 2). The resulting ionogels are named IG-f-x, where f and x are the mass ratio of the AA moieties in P(AA-co-AAm) and the mass fraction of the ionic liquid in the ionogel, respectively. Correspondingly, the FRCIs composed of IG-f-x and CF are named FRCI-f-x. The thickness of the FRCI is controlled to 1 mm. The mass ratio of the CF fabric in the FRCI is calculated to be 33.9% (Supplementary Fig. 2). PAA and [EMIM][OTf] can form a homogeneous and stretchable ionogel network because of their good compatibility (Supplementary Fig. 3a). On the other hand, the poor compatibility between PAAm and [EMIM][OTf] prompts the aggregation of amide groups to form hydrogen bonds during polymerization, leading to the ionogel becoming opaque (Supplementary Fig. 3b)[15]. Controlling the AA ratio between 0.8 and 0.95 can result in copolymers having good compatibility with [EMIM][OTf] (Supplementary Figs. 3c and 4). The infrared results indicate the presence of abundant non-covalent interactions between cations, anions, and polymer chains within the ionogel, including cation-oxygen interactions between imidazolium cations and carbonyl groups of AA moieties, as well as hydrogen bonding between anions and carboxyl groups. As shown in Supplementary Fig. 5, with increasing ionic liquid content, blueshifts are observed for $v$(C=O/PAA) and $v$(C−F), while redshifts occur for $v$(imidazole) and $v$(S=O). Additionally, when the ionic liquid content is fixed, as the AA content increases, the infrared peak trends align with Supplementary Fig. 5 due to the enhanced interaction between carboxyl groups and cations/anions (Supplementary Fig. 6). Meanwhile, altering both the ionic liquid content and AA content results in little change in the corresponding $v$(C=O/PAAm) and $v$(N−H) of PAAm, suggesting that the AAm moieties consistently exist in the form of hydrogen bonds within the ionogel and hardly interact with cations/anions (Supplementary Figs. 5 and 6). This is also consistent with the poor miscibility between PAAm and ionic liquid. Therefore, based on the above

analysis, we speculate that a 3D supramolecular network is formed between P(AA-co-AAm) and the ionic liquid (Fig. 1a). The rheological master curve reveals that the ionogel exhibits good thermal stability and remains in the rubbery state over a wide range of frequencies and temperatures (Supplementary Fig. 7a). By fitting the relationship between the shift factor and temperature (fitting temperature $>T_g + 50$ °C) using the Arrhenius equation, apparent activation energy of 59.3 kJ mol$^{-1}$ is obtained (Supplementary Fig. 7b). This indicates that the supramolecular network present in the ionogel possesses dynamic properties, which can dissipate energy through the dissociation of the supramolecular interactions during deformation. The glass transition temperature ($T_g$) of the ionogel gradually increases with the increase of the AAm ratio along with the enhancement of supramolecular networks (Supplementary Fig. 8). Therefore, ionogels with tunable mechanical properties can be obtained by modulating the supramolecular network within the ionogel through changing the ratio of the two monomers and the content of ionic liquids (Supplementary Fig. 9 and Supplementary Table 3).

The CF fabric is plain weave (Supplementary Fig. 10). The ionization of acidic groups on the fiber surface can make the surface negatively charged (Supplementary Fig. 11). The precursor solution of the ionogel can be well infiltrated into the CF bundles. As shown in Supplementary Fig. 12, the precursor solution exhibits good wettability with CF fabric, allowing it to rapidly penetrate into the interior of the fabric upon contact. After the polymerization, the soft and viscoelastic ionogel covers the surface of the CF fabric and fills in the interstices of the fiber bundles, forming a strong interlocking structure between CF and ionogel (Fig. 1b and Supplementary Fig. 13). The high strength and high modulus CF ensures that the FRCI can withstand large loads, while the strongly adherent ionogel guarantees that the force can be well dispersed to dissipate energy during the fracture process. Ultimately, the FRCI achieves high toughness and can even carry the weight of a human without crack extension (Fig. 1c).

### Mechanical properties

The FRCI possesses good mechanical properties because of the strong adhesion between the CF fabric and ionogels. Therefore, the adhesion properties between CF and ionogels were first evaluated. A single bundle of CF was embedded in the ionogel, and the critical length of the fiber bundle from fiber breakage to fiber pullout was determined by decreasing the embedding length of the fiber bundle in the ionogel. The interfacial bonding strength can be estimated by the following equation,

$$\tau_s = F/A \tag{1}$$

where $\tau_s$, $F$, and $A$ are the interfacial bonding strength, the force when the fiber pullout occurs, and the surface area of the fiber bundle that is embedded in the ionogel, respectively (Supplementary Fig. 14a). Taking IG-0.85-60% as an example, the interfacial bonding strength between IG-0.85-60% and fiber bundle is 4.8 MPa, which is much higher than the fracture strength of the ionogel (2.8 MPa) (Supplementary Fig. 14b). Therefore, during the deformation process, interfacial detachment between the ionogel and CF does not occur; instead, intense deformation of the ionogel primarily occurs to dissipate energy. Meanwhile, the strong adhesion between the ionogel and the CF bundle ensures that the adhesive force between them is greater than the breaking force of the CF bundle when stretched. Consequently, the high interfacial bonding strength can ensure the fiber bundles break rather than being pulled out when the bundles are embedded in different ionogels and kept at a depth of 10 mm, revealing the efficient interlocking structure (Fig. 2a). In addition, the lap-shear test indicates that there remains good adhesion between the CF fabric and the ionogel when the CF bundle is replaced with CF fabric (Supplementary Fig. 15).

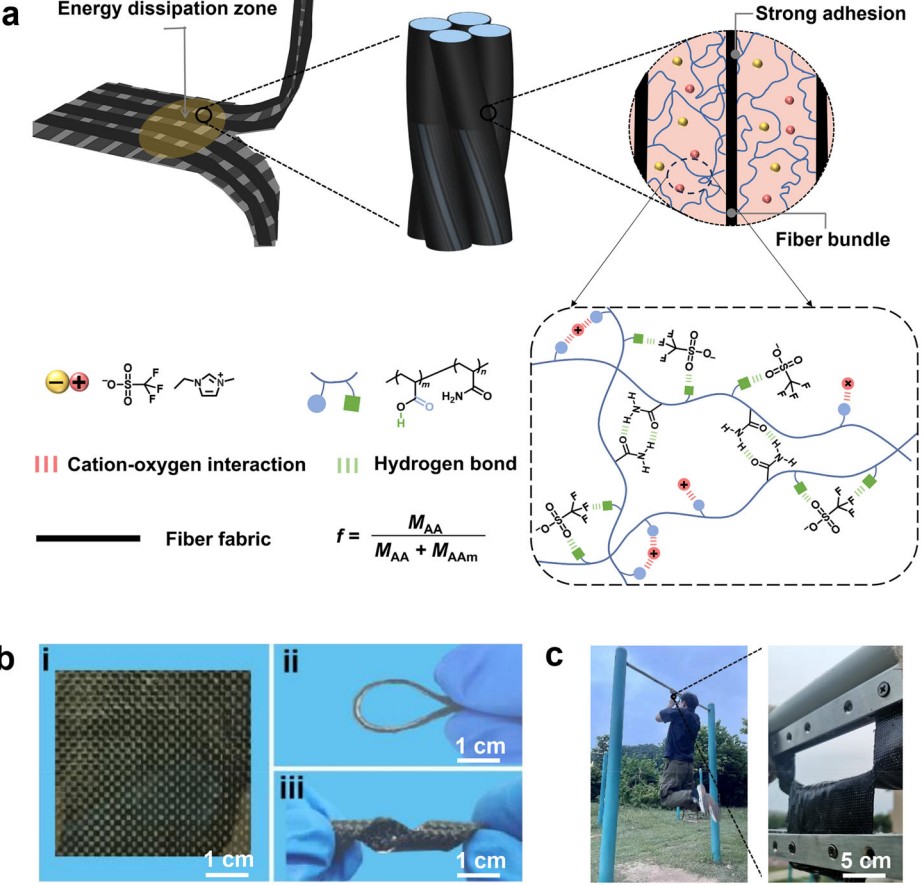

**Fig. 1 | Design of FRCI with high toughness and good deformability. a** Schematic of the FRCI composed of the CF fabric and supramolecular ionogel. The supramolecular ionogel is composed of polymer chains and ionic liquid through the cation-oxygen interactions and hydrogen bonding. **b** Photographs of the FRCI showing the bending and distorting processes. **c** Photographs of the FRCI showing crack resistance.

The load-bearing properties of the FRCI were measured by the uniaxial tensile test. IG-0.85-60% is soft and stretchable, with a tensile strength of 2.8 MPa and a work of extension of 12.2 MJ m$^{-3}$. On the contrary, the CF fabric has high strength (184 MPa) yet low stretchability, with an elongation at break and work of extension of 9.1% and 9.5 MJ m$^{-3}$, respectively. However, when the ionogel penetrates the CF fabric, the modulus of the resulting FRCI-0.85-60% is similar to that of the CF fabric, while its strength (315 MPa), stretchability (15.7%), and work of extension (29.1 MJ m$^{-3}$) are greatly enhanced (Fig. 2b). The mechanical properties remain stable when the proportion of monomers in the FRCI is changed (Supplementary Fig. 16a). However, its fracture strength can be further increased as the ionic liquid content decreases (Supplementary Fig. 16b). Furthermore, even when the temperature rises to higher levels, such as 50 °C and 80 °C, FRCI still maintains good load-bearing capacity. Compared to room temperature, the increase in temperature results in a slight decrease in the modulus and a slight increase in the stretchability (Supplementary Fig. 17). The improvement of the mechanical properties of FRCIs originates from the bridging effect of the ionogel between the CF bundles[29]. During the fracture process, the ionogel can transfer the force from the broken fibers to the adjacent fibers through shear deformation, i.e., the lost loading capacity due to fiber breakage can be made up by the fibers in the adjacent region. Thus, fiber breakage is not concentrated on a limited scale, and a larger overload region can be generated to delay material failure and increase the strength. Subsequently, we investigated the energy dissipation of FRCI during deformation through cyclic stress-strain curves (Supplementary Fig. 18a). Due to the tight bonding between the ionogel and CF fabric,

FRCI can dissipate a significant amount of energy during deformation, with the dissipated energy increasing as the strain increases. Across various strains, the damping capacity (the ratio of dissipated energy to input energy) can surpass 80% (Supplementary Fig. 18b). The high dissipated energy and damping capacity contribute to the resistance of FRCI to external impacts and tearing. In addition, the FRCI can still maintain high flexibility and bending capacity (Fig. 1b). Three-point bending experiments show that the bending modulus of FRCI-0.85-60% is 2.2 MPa (Fig. 2c and Supplementary Fig. 19). Compared to the tensile modulus (3.0 GPa), the bending modulus is three orders of magnitude lower. The changes in mechanical behavior and energy dissipation capacity of the FRCI after 5000 cycles were tested using the tensile mode of the dynamic mechanical analyzer. During the cycling process, both the modulus and loss factor of the FRCI remain stable (Supplementary Fig. 20).

The tearing toughness of FRCIs was then quantitatively measured by the trouser tearing test (Supplementary Fig. 21). Taking FRCI-0.85-60% as a representative, the effect of sample width ($w$) on the fracture behavior was first investigated. When the width of the FRCI is increased from 10 to 70 mm, the tearing toughness is greatly improved from 141 kJ m$^{-2}$ to 2278 kJ m$^{-2}$ (Supplementary Fig. 22). Then, the critical size for the FRCI to reach saturation toughness can be determined. The main resistance to prevent the fiber bundle from being pulled out of the FRCI during the fracture process comes from the shear stress between the ionogel and CF due to their strong adhesion. When the total shear stress applied to the fiber bundle by the ionogel is less than the fracture strength of the fiber bundle, the fiber bundle will be pulled out; conversely, the fiber bundle will fracture. Therefore, the fracture

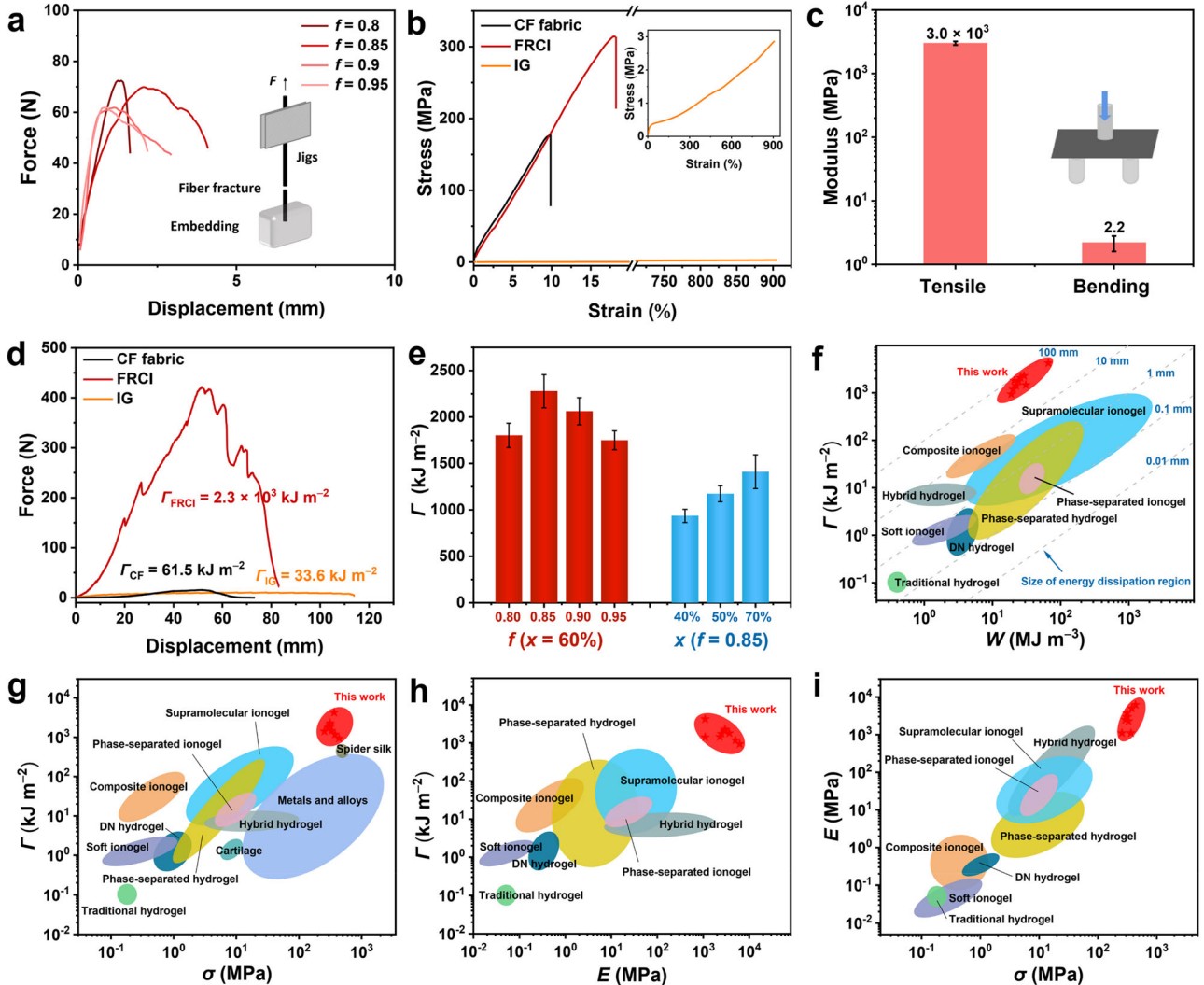

**Fig. 2 | Mechanical properties of FRCI. a** Fiber fracture force of the fiber bundle inside different ionogels ($x = 60\%$). **b** Tensile stress-strain curves of the CF fabric, ionogel, and FRCI-0.85-60%. **c** Tensile and bending modulus of FRCI-0.85-60%. **d** Force-displacement curves of the CF fabric, ionogel, and FRCI-0.85-60% in the trouser tearing test. **e** Tearing toughness of FRCIs with different AA ratios and ionic liquid contents. **f** Relationship between toughness and work of extension of FRCIs and other materials. **g–i** Comparison of FRCIs in this work with different materials in terms of toughness, tensile strength, and tensile modulus. Data in (**c** and **e**) are reported as their means ± SDs from $n = 3$ independent samples.

behavior and toughness of the FRCI change as its width increases. As shown in Supplementary Fig. 23, the morphologies of the FRCI reveal that with increasing $w$, the fracture behavior of the FRCI changes from the transverse fiber bundle being pulled out as the main fracture mode ($w \leq 30$ mm), to the coexistence of the pulling out and fracture of the fiber bundle ($w = 40$ mm), and to the fracture of the fiber bundle becoming the main fracture mode ($w \geq 50$ mm). The tearing toughness also increases with $w$, saturating at $w = 50$ mm of 2278 kJ m$^{-2}$ (Supplementary Fig. 24). When $w$ exceeds 50 mm, the toughness of the FRCI becomes independent of size and remains constant. Therefore, the widths of the samples were fixed to 50 mm to ensure that the fracture behavior of the samples was mainly fiber fracture and that no size-dependent effect would occur.

The tearing toughness of FRCI-0.85-60% (2278 kJ m$^{-2}$) is much greater than that of the CF fabric (61.5 kJ m$^{-2}$) and IG-0.85-60% (33.6 kJ m$^{-2}$), and even greater than their product (Fig. 2d). Moreover, the toughness of FRCIs can be modulated by changing the ratio of AA and the ionic liquid content since the mechanical properties of ionogels have a large scope for modulation (Fig. 2e, Supplementary Fig. 25 and Supplementary Table 4). Additionally, temperature variations do not affect the tearing toughness of FRCI, with the toughness at 50 °C

and 80 °C being almost identical to that at room temperature (Supplementary Fig. 26). The toughness is the product of the fractocohesive length and work of extension,

$$\Gamma = l_T \times W \qquad (2)$$

where $\Gamma$ represents the toughness to characterize the crack propagation resistance of a material, $l_T$ represents the fractocohesive length to characterize the size of the energy dissipation region, and $W$ represents the work of extension to characterize the dissipated energy density of a material before rupture. $l_T$ can be reflected by the slopes in the $\Gamma$-$W$ plot. Figure 2f reveals that the $l_T$ values of current gel materials (e.g., ionogels and hydrogels) are in the range of 0.01–10 mm, which fundamentally limits the crack resistance of the material (Supplementary Table 5)[8,13,15,18,20,23,30–38]. In contrast, our strategy can raise the $l_T$ value of composite ionogels to 50–100 mm, resulting in high toughness and crack insensitivity. This is also consistent with the saturation of the tearing toughness of FRCI at widths greater than 50 mm (Supplementary Fig. 24). Meanwhile, this result is also reflected in the fracture surface of the FRCI. The significant fracture of CF and the severe deformation of the ionogel indicate that the large energy

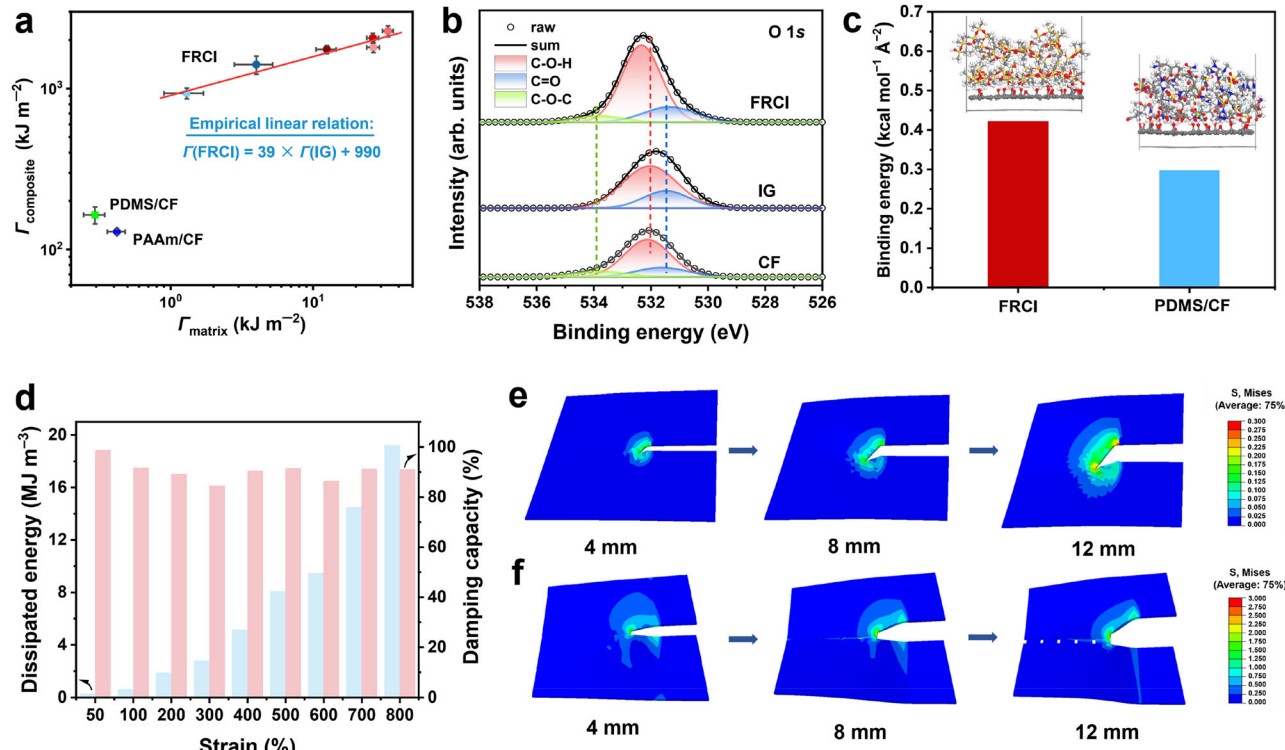

**Fig. 3 | Toughening mechanism of FRCI. a** Relationship between the tearing toughness of the composites and the tearing toughness of the polymer matrices. **b** XPS O 1*s* spectra of the FRCI, ionogel, and CF fabric. **c** Binding energies between components in FRCI and PDMS/CF. **d** Dissipated energy and damping capacity of the ionogel at different strains. **e**, **f** Stress nephograms of the FRCI (**e**) and PDMS/CF (**f**) during the fracture process. Data in (**a**) are reported as their means ± SDs from *n* = 3 independent samples.

dissipation region can dissipate a large amount of energy at the crack tip to resist the crack expansion (Supplementary Fig. 27a). Additionally, SEM results of the fracture surface also show that the strong interface between the ionogel and the CF fabric can enable them to remain tightly bonded without interfacial detachment after the intense shear deformation (Supplementary Fig. 27b, c). Meanwhile, FRCI also exhibits fatigue resistance with a fatigue threshold as high as 26.4 kJ m$^{-2}$ (Supplementary Fig. 28).

Therefore, the FRCI achieves an effective combination of different mechanical properties. The tensile strength, tearing toughness, and tensile modulus of the FRCIs are much higher than those of representative tough gel materials, such as double-network hydrogels, phase-separated hydrogels, hybrid hydrogels, supramolecular ionogels, composite ionogels, and phase-separated ionogels (Fig. 2g–i)[8,13,15,18,20,23,30–40]. The toughness is 100−2000 times higher than that of most reported ionogels and hydrogels because of the toughening effect of interfacial locking. More surprisingly, the toughness of the FRCI is much greater than that of the well-known high-performance spider silk in nature, and even two orders of magnitude higher than that of metals and alloys (Fig. 2g)[18,39,40]. Additionally, unlike conventional industrial materials, which are characterized by high strength but low toughness, the FRCI has both high strength and high toughness (Supplementary Fig. 29a)[25–27,41–46]. The relationship between specific strength and specific toughness further shows that the FRCI, which is located at the upper right of the plot, outperforms almost all of the current first-class industrial materials, suggesting that it perfectly combines the advantages of high toughness and low density (Supplementary Fig. 29b). The above results indicate that the FRCI not only has sky-high toughness, strength, and modulus but also breaks the contradiction between mechanical properties and density.

To better understand the role of strong interfacial locking between the ionogel and CF in enhancing the crack resistance of FRCIs,

polydimethylsiloxane (PDMS) elastomer/CF composite (PDMS/CF) and polyacrylamide (PAAm) hydrogel/CF composite (PAAm/CF) were synthesized for comparative analysis. PDMS is a neutral crosslinked elastomer, which can only be adhered to the CF fabric by weak van der Waals forces. The weak interfacial bonding of PDMS/CF makes it difficult to transfer the stress. Then, the fibers can only undergo interfacial detachment and be pulled out without fiber fracture (Supplementary Figs. 30 and 31), resulting in low toughness (164 kJ m$^{-2}$), which is only slightly higher than that of the CF fabric (61.5 kJ m$^{-2}$) (Supplementary Fig. 32). A similar result is also found in PAAm/CF with low toughness (129 kJ m$^{-2}$) (Supplementary Fig. 32). In contrast, the precursor solution of the ionogel can adhere around the fiber bundles and strong interfacial bonding can be formed during polymerization, which facilitates the energy dissipation of the FRCI. First, when the FRCI is subjected to a tearing force, the stress will be effectively dispersed through the shear deformation of the ionogel, generating a large energy dissipation region. Then, the bulk deformation and fracture of the tough ionogel also dissipate massive energy. Afterward, the fracture of CF can further release its stored elastic energy, leading to high toughness of the FRCI.

## Insights into the structure−property relationship

The relationship of toughness between the ionogel and FRCI further illustrates that their toughness follows the empirical linear equation (Fig. 3a and Supplementary Fig. 33),

$$\Gamma(\text{FRCI}) = 39 \times \Gamma(\text{IG}) + 990 \qquad (3)$$

The toughness of the FRCI increases by two orders of magnitude after toughening by CF. This result suggests that, in addition to the strong interfacial bonding between the ionogel and CF, the toughness

of the ionogel is one of the key factors in toughening the FRCI when fiber fracture occurs. The strong interfacial bonding between the ionogel and CF can effectively transfer the load from CF to the ionogel, thus realizing bulk energy dissipation of the ionogel and elastic energy release of CF. Therefore, FRCI-0.85-60% has the highest tearing toughness due to the highest tearing toughness of IG-0.85-60% among all ionogels. The intercept of 990 kJ m$^{-2}$ in Eq. 3 can be used to estimate the released energy of fractured fibers. However, when the interfacial bonding is weak, the fracture behavior is dominated by the fiber pullout process, preventing the release of elastic energy from CF. The deformation and fracture of the polymer matrix during fiber pullout become the main energy dissipation mechanism, resulting in a slight increase in toughness. Therefore, PDMS/CF and PAAm/CF are located much lower than the fitted curves of FRCI toughness (Fig. 3a).

To further illustrate the structure–property relationship of the FRCI, the interactions between the ionogel and CF were investigated. The X-ray photoelectron spectroscopy (XPS) results show that there is a tight bond between the ionogel and the CF fabric. As shown in Supplementary Fig. 34, the C 1$s$ peak of O–C=O shifts from 288.8 eV in CF and 288.5 eV in ionogel to 289.1 eV in FRCI, while the C 1$s$ peak of C–OH shifts from 285.7 eV in CF to 285.4 eV in FRCI. Meanwhile, the O 1$s$ peak of C=O in FRCI shifts to low binding energy while the O 1$s$ peak of C–OH in FRCI shifts to high binding energy (Fig. 3b). The changes in C 1$s$ peak and O 1$s$ peak indicate a change in the electron density of the carboxyl and hydroxy groups due to the formation of strong binding between the polymer chains and CF. In addition, the cations and anions of the ionic liquid will further strengthen the bond between the ionogel and the CF fabric. As illustrated in Supplementary Fig. 35, the N 1$s$ peak of the imidazole ring, as well as the S 2$p$ and F 1$s$ peaks of TFSI$^-$ in FRCI shift to higher binding energy compared to those in the ionogel, indicating the ionic interaction between the ionogel and CF. Thus, a 3D supramolecular network is initially formed between the ionic liquid and polymer chains, followed by the creation of a tight interface through the synergistic interactions among the polymer chains, ionic liquid, and CF fabric, collectively forming the FRCI. CT results also demonstrate that the ionogel within the FRCI is tightly bound to the carbon fibers, with no gaps present (Supplementary Fig. 36). The construction of interfacial binding in the FRCI can also be confirmed by molecular dynamics (MD) simulations and frequency-sweeping rheological experiments. The MD results show that the surface binding energy per unit area between the ionogel and CF is 0.422 kcal mol$^{-1}$ Å$^{-2}$, which is much stronger than that between neutral PDMS and CF (0.297 kcal mol$^{-1}$ Å$^{-2}$) (Fig. 3c and Supplementary Fig. 37). Meanwhile, the rheological results demonstrate that the loss modulus of the elastic ionogel rises rapidly once it is complexed with CF due to the bond breaking at the interface during deformation (Supplementary Fig. 38). Moreover, the loss factor of FRCI increases significantly with frequency, indicating enhanced energy dissipation capability, which further proves the tight interfacial bonding between the ionogel and CF fabric. In contrast, the rheological behavior of PDMS/CF is similar to that of elastic PDMS, indicating weak interaction between PDMS and CF and poor energy-dissipating ability (Supplementary Fig. 39). Structural information of polymers can be further obtained from their mechanical behavior through stress relaxation experiments. At the same initial strain, the FRCI maintains high applied stress for over 15 min, while PDMS/CF rapidly relaxes after 1 min (Supplementary Fig. 40). The difference in relaxation behavior further confirms the structural stability of the FRCI achieved by the good interfacial bonding between the ionogel and CF.

Besides the interaction between the ionogel and CF fabric, the ionogel also plays a significant role in enhancing the tearing resistance of FRCI by effectively dissipating energy through the dissociation of supramolecular interactions during deformation. We investigated the changes in internal interactions within the ionogel during deformation using infrared spectroscopy. The infrared results indicate that during deformation, the ionogel undergoes redshifts in the carboxyl v(C=O/PAA) as well as blueshifts in v(N–H), v(C–F) and v(S=O) due to the dissociation of cation-oxygen interactions and hydrogen bonds (Supplementary Fig. 41). Consequently, the energy dissipation of the ionogel during loading and unloading increases almost exponentially with strain, effectively dissipating energy during deformation (Fig. 3d and Supplementary Fig. 42). From a strain of 50–800%, the cyclic curves of the ionogel exhibit significant hysteresis, with the dissipated energy increasing from 0.28 MJ m$^{-3}$ to 19.2 MJ m$^{-3}$. Furthermore, at various strains, the damping capacity remains high (>85%), indicating that most of the input energy is used to dissociate supramolecular interactions for energy dissipation. Therefore, during deformation, the ionogel can also blunt crack propagation and inhibit the further spread of defects, exhibiting high tearing toughness (Supplementary Fig. 43).

Afterward, the stress distribution of the composites during the trouser tearing process was investigated by finite-element modeling (FEM) simulations to further explore the toughening mechanism of the FRCI. For the FRCI, stress can be transferred around the crack tip through the ionogel to reduce stress concentration (Fig. 3e). Moreover, the stress dispersion region in the FRCI can be further increased as the fracture proceeds. However, for PDMS/CF, the maximum stress is concentrated at the crack tip (Fig. 3f). In addition, the stress remains concentrated at the crack tip even though CF is gradually pulled out during the fracture process. Therefore, the stress at the crack tip in the FRCI is much more dispersed than that in PDMS/CF. As we mentioned above, this is mainly attributed to the strong interfacial interactions between the ionogel and CF, as well as the high toughness of the ionogel. Hence, the FRCI requires a higher energy to initiate its crack extension in the crack direction compared to PDMS/CF (Supplementary Fig. 44).

Based on the above analysis, the mechanism underlying the tearing resistance of FRCI is described as follows. The CF fabric can achieve strong interfacial bonding with the polymer chains and ions of the ionogel through synergistic effects. During the tearing process of FRCI, the tearing force initially causes significant deformation of the ionogel and distributes the force to a large energy-dissipating area at the crack tip. During tearing, the ionogel effectively dissipates energy through the dissociation of supramolecular interactions, and the dissipated energy increases rapidly with increasing deformation. When the tearing force exceeds a threshold value, the ionogel at the crack tip begins to rupture. Therefore, the tearing toughness of FRCI is positively correlated with the tearing toughness of the ionogel. Throughout this process, the tight bonding between the CF fabric and the ionogel prevents interfacial separation and avoids the extraction of CF bundles. The tearing force, dispersed across the entire energy-dissipating area, is transferred to the CF bundles, causing them to fracture and releasing the large amount of elastic energy stored in the CF, which resists crack propagation. Therefore, the synergy between CF fabric and the ionogel significantly enhances the tearing resistance of FRCI.

## Electrical properties and multi-functions

Along with good mechanical properties, FRCIs also have high ionic conductivity (Fig. 4a and Supplementary Fig. 45). Moreover, the FRCI has a highly sensitive electromechanical response and its resistance increases rapidly with the increase of strain, which can be used as a strain sensor to percept external stimuli and demonstrates its potential as a smart material. The GF values of the FRCI are 382.1 and 784.6 for small and large strains, respectively (Fig. 4b). The GF value and toughness of the FRCI are superior to those of currently reported flexible ion-conductive materials (Fig. 4c and Supplementary Table 6)[9,13,16,20,33–35,47–49]. In addition, the sensing signal remains sensitive over a wide temperature range (Supplementary Fig. 46). The increase in the GF value is most likely caused by the significant decrease in ionic conductivity due to the squeezing of the ionogel by the CF fabric

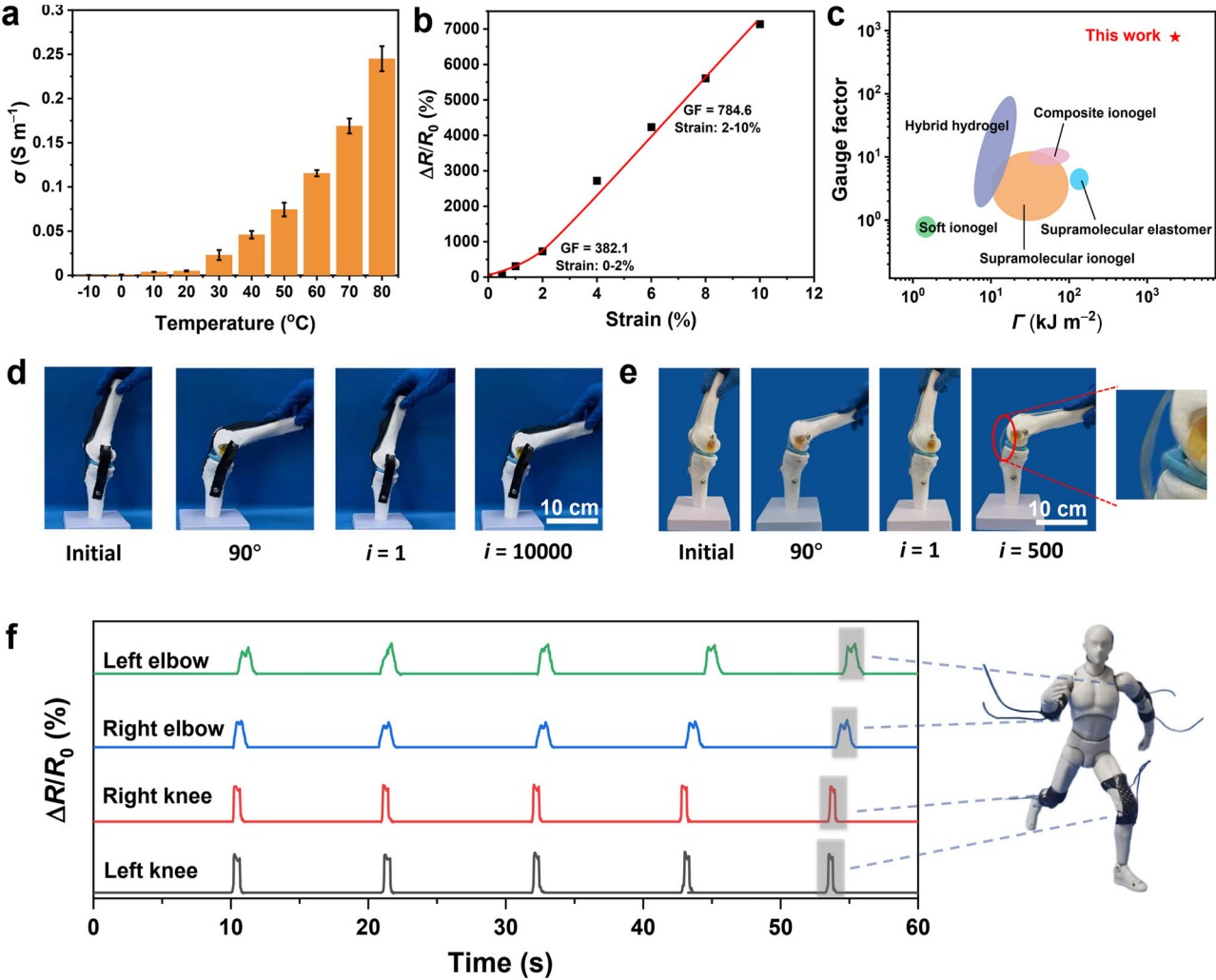

**Fig. 4 | FRCI with good ionic conductivity, sensitive sensing, and mechanical robustness. a** Ionic conductivities of FRCI-0.85-60% with increasing temperature. **b** GF values of FRCI-0.85-60% at different strains. **c** Comparison of GF values and toughness between this work and recently reported representative ion-conductive gels. **d**, **e** Photographs of the FRCI (**d**) and ionogel (**e**) fixing artificial bones. **f** Synchronized real-time resistance response of different robotic joints by the FRCI. Data in (**a**) are reported as their means ± SDs from $n = 3$ independent samples.

during deformation[47]. The FRCI also has a fast response time (Supplementary Fig. 47). Furthermore, the FRCI has good biocompatibility (Supplementary Fig. 48). Then, the movements of different joints can be reflected in real-time through the resistance change by directly adhering the FRCI to human elbows and knees (Supplementary Fig. 49). The FRCI can fix the artificial bones after bending 10,000 times, indicating their good durability and fatigue resistance (Fig. 4d). The stress-strain curve reveals that the strength and modulus of the composite ionogel only exhibit a slight decrease after bending 10,000 times (Supplementary Fig. 50). Moreover, the ionogel and CF fabric remain tightly bound together without separation (Supplementary Fig. 51). In addition, FRCI can still maintain high conductivity and sensitivity after 10,000 bending cycles, indicating that its conductive pathways remain uncompromised (Supplementary Fig. 52). In comparison, although the ionogel maintains its intact morphology due to its high stretchability, it shows gel softening after repetitive bending for 500 times (Fig. 4e). PDMS/CF suffers from fiber breakage after repeated bending for 1000 times due to its poor interfacial bonding ability and low toughness (Supplementary Fig. 53). Next, FRCIs adhere to the joints of a robot. The robust sensing performance of the FRCI is further confirmed by its ability to detect the synchronized real-time resistance response of different robotic joints (Fig. 4f). Additionally, even in the presence of cracks, FRCI can still safeguard artificial bones

from fracturing, unlike traditional elastomers which are prone to breaking (Supplementary Fig. 54). Meanwhile, regardless of whether cracks exist in FRCI, it can spontaneously perceive the timing and intensity of external impacts while providing impact protection, thereby enhancing reliability and extending service life (Supplementary Fig. 55a–c). In contrast, traditional ionogels are prone to damage when subjected to excessive impacts (Supplementary Fig. 55d). The above results indicate that the FRCI with good mechanical and sensing properties has great potential in wearable electronics, smart impact protection, and intelligent robots. Another point worth discussing is that current strain sensors primarily focus on low modulus flexible materials to enable mechanical matching and conformal contact with the epidermis during use. Therefore, the application of high-stiffness FRCI in epidermal strain sensors may be limited due to its lower elongation compared to biological skin, coupled with strengths and moduli that are much higher than skin. For FRCI, it may be more suitable for strain sensing in scenarios requiring high strength and high modulus, such as in robotic biomimetic ligaments.

## Universality of tough FRCIs
The design of high-performance FRCIs relies on the good interfacial bonding between ionogels and fabrics, as well as the high toughness of ionogels. To explore the generality of this strategy, different tough

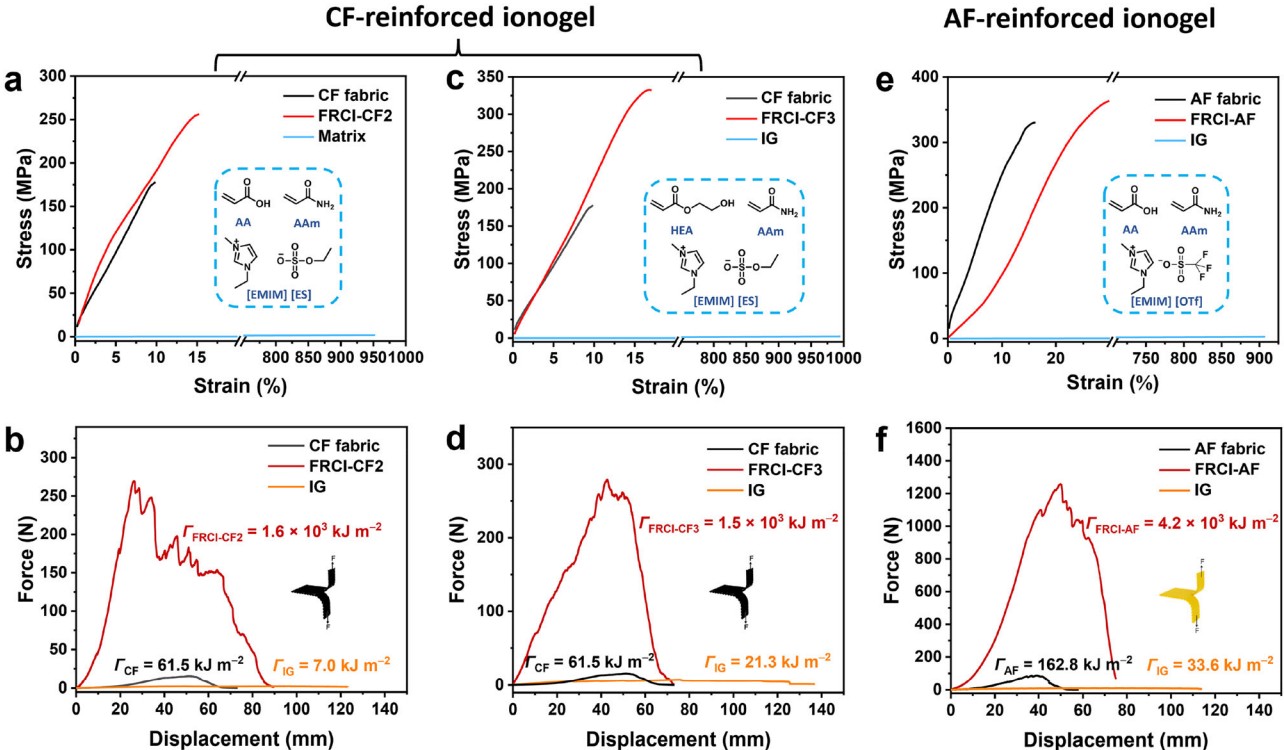

**Fig. 5 | Mechanical properties of FRCI-CF2, FRCI-CF3, and FRCI-AF. a** Tensile stress-strain curves of FRCI-CF2, CF fabric, and IG-2. **b** Force-displacement curves and toughness of FRCI-CF2, CF fabric, and IG-2 after tearing. **c** Tensile stress-strain curves of FRCI-CF3, CF fabric, and IG-3. **d** Force-displacement curves and toughness of FRCI-CF3, CF fabric, and IG-3 after tearing. **e** Tensile stress-strain curves of FRCI-AF, AF fabric, and IG-0.85-60%. **f** Force-displacement curves and toughness of FRCI-AF, AF fabric, and IG-0.85-60% after tearing.

ionogels were designed and complexed with different fabrics. The tough ionogel can be prepared by changing the types of monomer or the ionic liquid. IG-2 was generated by replacing the ionic liquid in IG-0.85-60% from [EMIM] [OTf] to 1-ethyl-3-methylimidazolium ethylsulfate ([EMIM] [ES]). IG-3 was further prepared by replacing the AA monomer in IG-2 with the hydroxyethyl acrylate (HEA) monomer. Then, FRCI-CF2 and FRCI-CF3 can be obtained by laminating IG-2 and IG-3, respectively, with CF fabrics, which show similar mechanical properties to FRCI-0.85-60%. Both FRCI-CF2 and FRCI-CF3 display high strength (257 and 332 MPa), high toughness (1618 and 1459 kJ m$^{-2}$), and low bending modulus (1.8 and 4.2 MPa) (Fig. 5a–d and Supplementary Figs. 56 and 57). Additionally, when the CF fabric in FRCI-0.85-60% is replaced with the aramid fiber (AF) fabric (Supplementary Fig. 58), the resulting FRCI-AF also exhibits good mechanical properties, with the strength of 365 MPa and work of extension of 66.1 MJ m$^{-3}$ (Fig. 5e and Supplementary Fig. 59). In particular, the tearing toughness reaches an impressive 4219 kJ m$^{-2}$ (Fig. 5f). Meanwhile, FRCI-AF still has good flexibility and low bending modulus (1.2 MPa) (Supplementary Fig. 59). Therefore, our synergistic toughening strategy is universal. Highly tough FRCIs can be customized by designing the type of ionogels, the type of fabrics, etc.

## Discussion

In summary, flexible FRCIs with high strength, high modulus, and good toughness can be obtained by incorporating high-performance fibers into supramolecular ionogels. The strong interfacial interactions between the fibers and ionogels can effectively disperse stress during tensile and fracture processes through the bridging effect of the ionogel. Consequently, the strength, modulus, and toughness of the FRCI can be synergistically enhanced while maintaining its flexibility, filling the gap between soft gel materials and traditional rigid materials. In addition, the ion-conducting properties of the FRCI

guarantee a fast response time and high sensitivity to deformation. Notably, due to its mechanical durability, the FRCI remains stable and undamaged after undergoing 10,000 bending cycles when fixing the artificial bone. Meanwhile, when used as an impact-resistant material, it can provide protection while simultaneously sensing the timing and intensity of impacts. This work presents a universal strategy to develop highly tough and multifunctional composite ionogels, which can facilitate the development of soft robots, smart fabrics, and wearable devices.

## Methods
### Materials

Plain weave carbon fiber (CF) fabric and aramid fiber (AF) fabric were purchased from Dongguan Yini Composites Co. Acrylic acid (AA, 99%), acrylamide (AAm, 99%), hydroxyethyl acrylate (HEA, 98%), 1-ethyl-3-methylimidazolium trifluoromethanesulfonate ([EMIM][OTf], 98%), 1-ethyl-3-methylimidazolium ethylsulfate ([EMIM][ES], 98%), and photoinitiator 2-hydroxy-2-methylpropiophenone (HMPP, 97%) were provided by Shanghai Aladdin Biochemical Science and Technology Co. Polydimethylsiloxane (PDMS) was provided by Hangzhou Westru Technology Co. All chemicals were used as received without further purification.

### Sample preparation

The preparation process is shown in Supplementary Fig. 1. Taking IG-0.85-60% and FRCI-0.85-60% as an example, AA (3.4 g), AAm (0.6 g), and HMPP (9.0 μL) were mixed with [EMIM][OTf] (6.0 g) to obtain a well-dispersed solution. The precursor solution was injected into a mold and polymerized under UV light (365 nm) irradiation for 24 h to obtain the ionogel and FRCI. Other ionogels and FRCIs were prepared in a similar manner as described above, by varying the types of fabrics, monomers, and ionic liquids.

## Characterization

A Supra 55 equipment was used to acquire scanning electron microscope (SEM) micrographs with an acceleration voltage of 10 kV after coating the samples with gold. X-ray photoelectron spectroscopy (XPS) results were obtained on Thermo Scientific K-Alpha equipment. Zeta potential was measured by a Zeta potential analyzer (Anton Paar Surpass 3). Wide-angle X-ray scattering and small-angle X-ray scattering experiments were conducted using the Ganesha SAXS Lab system. The rheological and stress relaxation experiments were carried out using a rheometer (Thermo-Fisher Mars40). The data were analyzed using OriginPro 2025, ver 10.2.

## Uniaxial tensile test

The uniaxial tensile tests of ionogels, hydrogels, and PDMS were carried out in an electronic universal testing machine (Xiamen Meters Instruments) equipped with a 100 N load cell. The uniaxial tensile tests of composites and fabrics were conducted on an electronic universal testing machine (CMT5504) equipped with a 10 kN load cell. Rectangular specimens (80 mm in length and 10 mm in width) were prepared with the fiber bundles parallel or perpendicular to the length direction. The gauge length of the specimen was 20 mm. The fracture strength was determined by dividing the maximum load by the cross-sectional area of the sample (the width, $w$, multiplied by the thickness, $t$). The work of extension was defined as the area under the stress-strain curve.

## Trouser tearing test

The tearing toughness of the specimens was determined by the trouser tearing test. The experiment was performed on an electronic universal testing machine (CMT5504) equipped with a 10 kN load cell. The width of the trouser tearing sample was $w$, $L_{bulk}$ was $w/2 + 5$ mm, and there was an initial notch of 20 mm in the middle of the sample along the length direction. The stiff tapes were stuck to the legs of the ionogel to avoid being elongated. The tensile speed was 50 mm min⁻¹. The tearing toughness was calculated from the tearing force-displacement curve using the following equation.

$$\Gamma = \frac{\int_0^L F dL}{t \cdot L_{bulk}} \quad (4)$$

where $F$ was the tearing force, $t$ was the thickness of the specimen, $L$ was the displacement, and $L_{bulk}$ was the length of the crack.

## Three-point bending test

Three-point bending test was used to characterize the flexibility of the FRCI on an electronic universal testing machine (GC2022020) equipped with a 100 N load cell. The length and width of the specimen were 70 mm and 20 mm, respectively, and the length between the bottom points was 40 mm. The testing rate was 10 mm min⁻¹. The bending modulus was calculated from the bending stress-strain curve. The bending stress was calculated as:

$$\sigma_f = \frac{3PL}{2bt^2} \quad (5)$$

The bending strain was calculated as:

$$\epsilon_f = \frac{6\delta t}{L^2} \quad (6)$$

where $P$ was the applied load, $L$ was the length between the bottom points, $b$ was the width of the specimen, $t$ was the thickness of the specimen, and $\delta$ was the displacement.

## Lap-shear test

The ionogel precursor solution was brushed between two pieces of CF fabric to form a bonded area of $1 \times 1$ cm², followed by in situ polymerization. After the polymerization was completed, a universal testing machine was used to conduct tensile testing under ambient conditions at a speed of 50 mm min⁻¹. The adhesion strength was calculated through the force divided by the contact area.

## Molecular dynamics simulation

Material Studio was used to perform the modeling and simulation processes in the molecular dynamics simulation. A periodic model of P(AA-co-AAM)/[EMIM][OTf] ionogel containing 10 P(AA-co-AAM) chains (1 repeating unit of AAm and 5 repeating units of AA in one chain) and 17 ionic liquid molecules was constructed in an amorphous cell (box size: $24.000 \times 29.800 \times 14.157$ Å³). Additionally, a periodic model of PDMS elastomer containing 10 polymer chains (8 repeating units in one chain) was constructed in an amorphous cell (box size: $24.000 \times 29.800 \times 15.769$ Å³). Dmol3 module was used to perform structure optimization and charge assignment of the molecules. The amorphous cell module was used for the amorphous modeling of the molecular structure. Then, the build layer tool was used to cover the amorphous cell onto the surface of CF with an initial distance of 4 Å. The Forcite module was used to perform the initial structural optimization and molecular dynamics simulations. The simulations were carried out under the NVT system with a simulation time of 0.5 ns. ESP charges were used for the charges, the simulation temperature was set to 298 K, and the compass force field was used as a description of the potential function to carry out the simulations. Finally, the binding energy was calculated using the following equation:

$$E_{Bind} = (E_{total} - E_C - E_{Cover})/S \quad (7)$$

where $E_{bind}$, $E_{total}$, $E_c$, $E_{cover}$, and $S$ represented the binding energy, the total energy, the energy of the CF surface, the energy of the elastomer on the CF surface, and the contact area, respectively.

## Finite-element modeling

The tearing process of the FRCI and PDMS/CF was simulated using ABAQUS software. The size of the model was $50 * 50 * 0.25$ mm³ with a 20 mm notch on one side edge for simulating the displacement loading. The tearing path was 30 mm. The CF fabrics and polymer matrices in the FRCI and PDMS/CF were simulated separately, and the fiber bundle and matrix were connected by a cohesive unit. As the displacement increased, the crack in the FRCI started to expand from the notch, while the fiber bundle in PDMS/CF was withdrawn from the matrix. The models were loaded in the same way as the test procedure. Based on the simulation results, the stress distribution of the model can be extracted.

## Electrical properties

The electronic properties of the FRCI were carried out using an electrochemical workstation (CHI660e, CH Instruments). The FRCI was cut into a circle with a diameter of 16 mm and sandwiched between two stainless steel electrodes. AC impedance measurements were performed with a frequency range of 1 MHz to 1 Hz and an amplitude of 5 mV. The conductivity of the FRCI was calculated using the equation:

$$\sigma = \frac{L}{R \times S} \quad (8)$$

where $R$, $S$, and $L$ were the resistance, area of the FRCI, and distance between the two electrodes, respectively. The sensing performance was obtained using a Keithley source meter (2612B), and the real-time resistance of the FRCI was recorded. The gauge factor (GF) was defined as

$$GF = \frac{\Delta R/R_0}{\varepsilon} \quad (9)$$

where $\varepsilon$ was the applied strain and $\Delta R/R_0$ was the change in resistance. Body movements were monitored by placing the FRCI on different joints.

### Cytotoxicity test

Mouse corneal epithelial cells (purchased from Procell, Cat NO. CP-M120) were incubated with different amounts of FRCI-0.85-60%, and the resulting samples were then tested for toxicity using the MTT method. Initially, FRCI-0.85-60% was sterilized under UV light for a duration of 24 h. Subsequently, sterilized FRCI-0.85-60% at concentrations of 1.0 mg mL$^{-1}$ was immersed in HCE-T special medium for a period of 24 h at 37 °C to obtain the leachate. Corneal epithelial cells were then inoculated in 96-well plates at a density of $4 \times 10^3$ cells per well and cultured for 24 h at 37 °C in a 5% CO$_2$ environment. Subsequently, different proportions (the ratio of leachate to culture fluid) of leachate were added and incubated for 24 h. After that, 20 μL of MMT solution (prepared as 5 mg mL$^{-1}$ with PBS buffer) was added to each well and incubated for 4 h. The absorbances were measured at 450 nm using an enzyme meter.

### Reporting summary

Further information on research design is available in the Nature Portfolio Reporting Summary linked to this article.

## Data availability

The data that support the findings of this study are available from the corresponding author upon request. The data generated in this study are provided in the article and the Supplementary Information. Source data are provided with this paper.

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

## Acknowledgements

X.L. acknowledges the financial support from the National Natural Science Foundation of China (22203015), Fujian Science & Technology Innovation Laboratory for Optoelectronic Information of China (2021ZZ127), and National Key Research and Development Program of China (2020YFA0710303). P.Z. also acknowledges the financial support from Funding for Top Hospital and Specialty Excellence of Fujian Province (2022(884)), Fujian Provincial Natural Science Foundation of China (2024J01626), and Fujian Provincial Health Technology Project (2024QNA016).

## Author contributions

X.L., K.Y., and H.Z. carried out experiments. X.L., P.Z., and Z.Z. supervised the project. X.L. and P.Z. co-wrote the manuscript. Z.S. assisted in the conduct of experiments as well as the confirmation of supramolecular structure. All authors discussed the results and revised the manuscript.

## Competing interests

The authors declare no competing interests.
