## [Transparent Peer Review file · Nature Communications]

Tough Fiber-Reinforced Composite Ionogels with Crack Resistance Surpassing Metals

Corresponding Author: Professor Xiaolin Lyu

Version 0:

Reviewer comments:

Reviewer #1

(Remarks to the Author)

Inspired by the structure-property of ligament, this work develops the carbon fiber reinforced ionogels, where the polymer matrix is tightly linked with carbon fiber via electrostatic interactions. The mechanical tests indicate that the achieved composite ionogels demonstrate high mechanical performance in terms of high strength, elastic modulus and tearing resistance. The authors further explore the application as strain sensor. Although the authors conduct many macroscopic mechanical properties and benchmark fracture metric with previous references, the underlying mechanism towards enhanced tearing resistance is not well analyzed, and there are many issues needed to be considered. Moreover, the claimed metric advancements are not convincing compared to a recent similar work in Adv Mater (Adv. Mater. 2024, 36, 2406252), so I do not think this work present strong innovation for publication in Nat. Commun. Here are some critical defects.

1. The comprehensive component interactions/cross-linking structure is missing, which makes the authors difficult to understand the correlation between structure and performance. Moreover, it looks there exists abundant reversible interaction between ionogel matrix and carbon fiber reinforce phase, and considering the focus of this work is the fracture toughness, the fundamental energy dissipation are definitely needed, unfortunately, the authors only examine the monotonic loading profile.
2. The fundamental mechanism for how carbon fiber reinforces ionogel via charge interaction is not well demystify, where the authors only conduct a series of macro-level mechanical performance. That is, the manuscript lacks dedicated in-depth analysis which is necessary for publication in a journal like Nat. Commun.
3. For the application section in Fig.4b, it obviously belongs to long-term cyclic test, so the anti-fatigue related properties must be considered in the previous mechanics section (Fig.2, 3).
4. Demonstration in Supplementary Figure 33 and sensing section in Fig.4 is not convincing. Are the claimed mechanical performance combinations (strength, modulus, and toughness) indispensable for such kind of application? The previous 3 figures largely deal with mechanics, why it changes to electrical performance? Without these rationales, these results are just a stacking of data and images.
5. The ideal of this work is much like a previous work ((Adv. Mater. 2024, 36, 2406252)), where the carbon fiber was used to reinforce polyurethane, and the reported mechanical metrics do not truly outperform the existing one. Also, in that reference, the underlying tearing resistance mechanism has been well demystified. In this context, the current work does not bring more new insight over previous one.

Based on the above limitations in terms of experiment designing, scientific novelty, and practical specialty. Overall, this work is uninspiring and I do not support its publication.

Reviewer #2

(Remarks to the Author)

In this manuscript, Lyu and co-workers have reported a tough fiber-reinforced ionogels by incorporating high-performance fibers into elastic ionogels to efficiently dissipate energy. It exhibited extraordinary tearing toughness (4219 kJ m⁻²), remarkable strength (365 MPa), high elastic modulus (1.0 GPa), and low bending modulus (2.2 MPa). Due to good ionic conductivity, fast response time, and exceptional strain sensitivity, the FRIG demonstrated great potential in intelligent robots and wearable devices. The following issues should be addressed before its acceptance:

1. Some similar works have been previously described in another article published in *Advanced Material* (please refer to *Adv. Mater.* 2020, 32, 1907180; *Adv. Mater.* 2024, 36, 2406252). In this case, the authors have to further claim their advances compared with these works in Introduction.
2. The collagen fiber bundle in the tendon is anisotropic, whereas the carbon fiber in this paper is isotropic. The strengthening principles of these two materials are not compatible, so the authors may need to redefine them.
3. The thickness and fiber content of the composite ionogel should be provided.
4. What is the wettability of carbon fiber and premixed solution (the blend of ionic liquids and monomers)?
5. The authors need to provide CT images to observe the internal structure of the FRIG.
6. In Figure S14, there is not a significant transition from being pulled to fracture of the carbon fiber from 30 to 70 mm. The authors should show this transition in a more intuitive way.
7. The strongly adherent between carbon fiber and ionic gel has an important effect on the mechanical properties of FRIG. It is necessary to provide a interlayer shear test for more detailed verification, and the corresponding standard test method should be described in detail.
8. In Figure 2e, the authors should provide an explanation on the maximum tearing toughness while $x = 60\%$, $f = 0.85$.
9. In Figure 2e and S10, why does the proportion of polymer monomers have little effect on tensile strength and elongation at break, while having a greater effect on trouser tearing?
10. In Figure 3b, the results of XPS only prove that carbon fibers interact with PAA and PAAm. The authors should provide evidence to prove the interaction between ionic liquid and carbon fiber.
11. In Figure 3d, the relationship between damping capability and interfacial bonding should be further explained.
12. In Figure 4, what is the conductivity and sensitivity of the FRIG after 10,000 bending cycles, the author should provide the corresponding experimental data.
13. After 10,000 cycles of testing under stretching conditions, whether the carbon fiber and ionogel are separated at the interface?
14. There are some mistakes in this manuscript in line 66, line 117, and line 234, the authors should check the full text carefully.

Reviewer #3

(Remarks to the Author)

The authors have developed tough fiber-reinforced ionogels with notable crack resistance by incorporating high-performance fibers into elastic ionogels. However, the novelty of this strategy needs to be highly improved.

1. On page 2, lines 21-23, the authors claim that fiber-reinforced ionogels can efficiently dissipate energy. However, there is no supporting data in the manuscript to substantiate this conclusion, such as cyclic loading-unloading tests.
2. Several images in the manuscript lack a scale bar, including Figure 1d, Figure 1e, Figure 4d, and Supplementary Figure 2, which compromises the clarity and reproducibility of the presented data.
3. On page 7, lines 119-136, the manuscript states that the strength of CF fabric (184 MPa) is greater than the interfacial bonding strength (4.8 MPa) and the strength of the ionogel (2.8 MPa). This contradicts the assertion that "the fiber bundles break rather than being pulled out."
4. On page 8, lines 136-140, it is mentioned that the strength of FRIG-0.85-60% increased to 315 MPa. A more detailed explanation of the mechanism behind the fiber-reinforced ionogel would be beneficial, as the strength of the ionogel may predominantly derive from the carbon fiber.
5. The manuscript lacks information on the mass ratio of carbon fiber to ionogel. Including this data would provide a better understanding of the primary factors contributing to the high strength of the material.

Version 1:

Reviewer comments:

Reviewer #1

(Remarks to the Author)

For finite-element modeling, there are some notable yellow spots in Fig.3e, corresponding to the higher stress concentration, whereas there is overall blue one in the PDMS/CF (3f), this analysis is contrast to the experimental result (the stress at the crack tip in the FRIG is much more dispersed than that in PDMS/CF.). Besides, there are some typos (page 16, there are no fig. 3 g and 3h).

'Moreover, the FRIG has a highly sensitive mechanical responsiveness and its resistance increases rapidly with the increase of strain, which can be used as a strain sensor to percept external stimuli and demonstrates its potential as a smart material.' FRIG shows high crack resistance with large energy dissipation, could the authors please explain how to

understand the term of 'mechanical responsiveness'

For strain sensor section, some preliminary biocompatibility/cell toxicity test should be considered if they develop its application as artificial bones. The authors examined the temperature dependent conductivity, so how about the sensing signal over the such temperature range. Besides, the authors highlight its excellent GF over previous references, while the superficial application in Fig.4f by only installing on a robot toy cannot reflect this metric. Moreover, just mentioned in initial comment, the strain scenarios in Fig.4f and Fig.S44 does not require the sensors possessing the high crack resistance. The authors must consider this point well.

For the fatigue resistance, some basic parameters, such as fatigue threshold, must be considered. More specific parameters should be examined after long-term loading cycles.

Considering such a high fiber content (34 wt%), it is doubtful to define it as ionogels (elastomer/composite ionogel may be more suitable).

The authors may also consider the temperature dependent mechanical performance over a wide range.

Page 2 'The inherent fragility of supramolecular bonds also poses a limitation on the strength and toughness', in fact, there are abundant references about tough and strong elastomers based on supramolecular bonds.

The stress value for FRIG does not agree with each other, where it reaches to 120 MPa in loading-unloading curves (Fig.S17), whereas it is only about 90 MPa in Fig.2b.

In the reply 'Moreover, after multiple cycles, the yielding of these flexible soft materials leads to softening, making it difficult to effectively fix joints and perform sensing and monitoring, as demonstrated in Fig. 4e.' The authors fail to prove the specific mechanics of the FRIG after cyclic loading. In addition, 'our FRIG represents a first universal example of toughening ionogels with high-performance fiber fabrics, providing new insights into the design of toughened flexible conductive gel materials.' It should be noted that the prototype of mechanical reinforcement for gel using fabric matrix can be traced by to Gong et al in 2016 (<https://doi.org/10.1002/adfm.201605350>, <https://doi.org/10.1039/C5MH00127G>)

According to maximal strain ratio for FRIG is less than 20% (Fig.S16 a, b), whereas for the body motion strain sensor in Fig.4, normally the tolerant strain range for body joint must be large than 30%, this result indicates that the achieved FRIG may not suitable for body motion monitor.

Reviewer #2

(Remarks to the Author)

The authors have well addressed my concerns. In addition, I also evaluated the responses of the authors to the concerns from Reviewer #3. All the issues raised by Reviewer #3 have been addressed, including the strategy novelty, dissipation energy, the clarity of images, error bars, and the mechanism behind the fiber-reinforced ionogel. Therefore, the current version is recommended for publication.

Version 2:

Reviewer comments:

Reviewer #1

(Remarks to the Author)

1. For the strain sensor application on human skin, the Young's modulus and bending stiffness should match the modulus of human stratum corneum (~150 kPa), and closely follow the skin under various deformation with robust and reliable attachment between the sensor and skin.

On average, biological skin is stretchable to 75% strain, and this allows free movement of the joints, which experience surface strains up to 55% for the knees (Bao et al. Pursuing prosthetic electronic skin. *Nature Mater* 15, 937–950 (2016)). In this work, the Young's modulus is even up to 103-104 MPa in Fig.2i and extremely high bending modulus. Moreover, another point that bothers my understanding is how the FRIG was attached to the knee and elbow (Fig.S49) and artificial bone (Fig.4d) and the authors fail to prove the conformal contact with the epidermis in a manner that does not constrain or alter natural motion or behaviors. Given the extremely high stiffness (and may lack of conformal contact with skin, the authors do not release this information at current stage), I do not think this FRIG is suitable for strain sensor even if they alter the items of ionogel and fabrics in Fig.5.

2. in the author reply, 'Therefore, this was the main point we intended to convey in our previous reply, namely that we pioneered the toughening of ionogels using fabrics, achieving significantly stronger performance compared to hydrogels, and providing new insights into the design of toughened ionogel materials.' I think the authors overestimate the potential of their ideal of fabric reinforced ionogels and should tone down this claim. In fact, there are other similar studies published. For example, Huang et al reported a mechanically interlocked ionic gel–elastomer (thermoplastic polyurethanes) hybrid material as soft strain gauge a tiny detection limit of 0.05% strain, ultrafast time resolution of 0.495 ms, and high linearity (<https://advanced.onlinelibrary.wiley.com/doi/10.1002/adv.202301116>). Similarly, Zhang et al applied a direct-ink-write 3D

printing process to produce a self-healing and rigid skeleton, and then an ionogel acting soft matrix was injected into the 3D skeleton, and they also specifically examined the resistance to crack growth (<https://advanced.onlinelibrary.wiley.com/doi/10.1002/adma.202405776>). Alternatively, Zhang et al reported a thermoplastic polyurethane nanonet-supported ionogel sensor (<https://advanced.onlinelibrary.wiley.com/doi/10.1002/adfm.202415694>). Also, Jiang et al reported a tough and fatigue-resistant ionic elastomer through the interlocking of a thermoplastic polyurethane (TPU) fibrous scaffold and an ionic supramolecular biopolymer matrix (<https://onlinelibrary.wiley.com/doi/10.1002/anie.202411418>). In this context, as it appears to me that the major unique selling point (reinforcement fabric + soft ionogel) of the present manuscript is incremental, and thus the novelty of the research is questionable, plus the strain sensor application scenario is not suitable due to the extremely high stiffness and may also lack of conformal contact with body, which have been noted above. Although some mechanical performance metric shows their advantage in Fig.2, this work does not adequate to make it innovative enough considering the high standbars of Nat Commun.

3. A minor confusion is they relate the high GF to the significant decrease ionic conductivity over a wide temperature range (page 17), whereas according to the results in Fig.4a, the conductivity increases sharply with increasing of temperature.

Response to Reviewers

We are appreciative of the feedback from our peer reviewers. Their constructive comments have significantly contributed to enhancing our manuscript. In response to their concerns, we have expanded our research with additional experiments, analyses, and discussions. These improvements are thoroughly integrated into the revised manuscript.

To facilitate a clear understanding of our revisions, we have organized our responses as follows:

Reviewer Comments and Our Responses: Each comment from the Reviewers (presented in black text) is followed by our corresponding response (in blue text).

Highlighted Text in the Revised Manuscript: Text modifications in the revised manuscript are red in the response letter.

Response to the Reviewer #1

Comment: Inspired by the structure-property of ligament, this work develops the carbon fiber reinforced ionogels, where the polymer matrix is tightly linked with carbon fiber via electrostatic interactions. The mechanical tests indicate that the achieved composite ionogels demonstrate high mechanical performance in terms of high strength, elastic modulus and tearing resistance. The authors further explore the application as strain sensor. Although the authors conduct many macroscopic mechanical properties and benchmark fracture metric with previous references, the underlying mechanism towards enhanced tearing resistance is not well analyzed, and there are many issues needed to be considered. Moreover, the claimed metric advancements are not convincing compared to a recent similar work in Adv Mater (Adv. Mater. 2024, 36, 2406252), so I do not think this work present strong innovation for publication in Nat. Commun. Here are some critical defects.

Reply: We are deeply grateful to the Reviewer for the comments on our manuscript. These invaluable suggestions have significantly contributed to enhancing the exploration of the structure-property relationship and toughening mechanism. Therefore, in *Answer 1*, we have conducted additional experiments and structural analysis to delve deeper into the interactions and crosslinking structures among the FRIG. Furthermore, in *Answer 2*, we have supplemented our experimental data to further discuss the structure-property relationship of FRIG, combining microscale

interaction analysis with macroscale mechanical behavior characterization to provide a refined analysis of its toughening mechanism. Additionally, we have elaborated on the significant implications of the studied mechanical property combinations for practical applications within our article. More importantly, in *Answer 5*, we have detailed the distinctions, advancements, and innovations of our revised manuscript compared to the literature (*Adv. Mater.* **2024**, *36*, 2406252), covering aspects such as material design, performance, toughening mechanism, and applied research (Please refer to *Answer 5* for details).

In summary, we are immensely thankful for the Reviewer's comments and suggestions. We kindly request you to review our revised manuscript based on our responses. Thank you once again for your time and invaluable feedback!

Question 1: The comprehensive component interactions/cross-linking structure is missing, which makes the authors difficult to understand the correlation between structure and performance. Moreover, it looks there exists abundant reversible interaction between ionogel matrix and carbon fiber reinforce phase, and considering the focus of this work is the fracture toughness, the fundamental energy dissipation are definitely needed, unfortunately, the authors only examine the monotonic loading profile.

Answer 1: Thanks for the Reviewer's suggestion. Your suggestion to elucidate the component interactions and cross-linking structures is important for clarifying the structure-performance relationship of FRIG. Therefore, we supplemented a detailed study.

1. Internal interactions within the ionogel: Infrared spectroscopy reveals that the ionogel primarily forms a 3D supramolecular network through hydrogen bonding and cation-oxygen interactions (Supplementary Fig. 5 and 6). Simultaneously, the time-temperature superposition master curve from rheological analysis demonstrates that the supramolecular network exhibits excellent dynamic properties, with an apparent activation energy of 59.3 kJ mol^{-1} , enabling energy dissipation through the dissociation of supramolecular interactions during deformation (Supplementary Fig. 7). These results and corresponding discussions are now reflected in the revised manuscript and revised supplementary information.

Supplementary Figure 5. Infrared results of P(AA-co-AAm), IL, and ionogels with the AA ratio of 0.85 and different IL contents. (a) $\nu(\text{C=O/PAA})$ and $\nu(\text{C=O/PAAm})$; (b) $\nu(\text{N-H})$; (c) $\nu(\text{imidazole ring})$; (d) $\nu(\text{C-F})$ and $\nu(\text{S=O})$.

Supplementary Figure 6. Infrared results of P(AA-co-AAm), IL, and ionogels with the IL content of 60% and different AA ratios. (a) $\nu(\text{C=O/PAA})$ and $\nu(\text{C=O/PAAm})$; (b) $\nu(\text{N-H})$; (c) $\nu(\text{imidazole ring})$; (d) $\nu(\text{C-F})$ and $\nu(\text{S=O})$.

Supplementary Figure 7. (a) Rheological master curve of IG-0.85-60% from 5 °C to 145 °C. (b) Relationship between the shift factor and temperature.

2. Cross-linking structure between the ionogel and CF fabric: XPS results also provide additional insights, showing that CF fabric can simultaneously bond with both the polymer chains and ionic liquids in the ionogel, leading to exceptionally strong interfacial interlocking (Fig. 3b and Supplementary Fig. 31 and 32). Consequently, as observed in CT images, the ionogel and CF fabric in FRIG are tightly bound together without any gaps (Supplementary Fig. 33). These results and corresponding discussions are now reflected in the revised manuscript and revised supplementary information.

Figure 3b. XPS O 1s spectra of the FRIG, ionogel, and CF fabric.

Supplementary Figure 31. XPS C 1s spectra of the FRIG, ionogel, and CF fabric.

Supplementary Figure 32. XPS N 1s (a), S 2p (b), F 1s (c) spectra of the FRIG, ionogel, and CF fabric.

Supplementary Figure 33. CT results of the FRIG. Scale bar: 500 μm .

3. Energy dissipation experiments: We conducted a study on energy dissipation using cyclic stress-strain curves (Supplementary Fig. 17). The results reveal that the FRIG can dissipate a significant amount of energy during deformation, with the dissipated energy increasing as the strain increases. High energy dissipation aids FRIG in effectively resisting external impacts and tearing.

Supplementary Figure 17. (a) Cyclic stress-strain curves of FRIG at different strains; (b) Corresponding dissipated energy and damping capacity at different strains.

We supplement the discussion on Page 5 of the revised manuscript.

“The infrared results indicate the presence of abundant non-covalent interactions

between cations, anions, and polymer chains within the ionogel, including cation-oxygen interactions between imidazolium cations and carbonyl groups of AA moieties, as well as hydrogen bonding between anions and carboxyl groups. As shown in Supplementary Fig. 5, with increasing ionic liquid content, blueshifts are observed for $\nu(\text{C=O/PAA})$ and $\nu(\text{C-F})$, while redshifts occur for $\nu(\text{imidazole})$ and $\nu(\text{S=O})$. Additionally, when the ionic liquid content is fixed, as the AA content increases, the infrared peak trends align with Supplementary Fig. 5 due to the enhanced interaction between carboxyl groups and cations/anions (Supplementary Fig. 6). Meanwhile, altering both the ionic liquid content and AA content results in little change in the corresponding $\nu(\text{C=O/PAAm})$ and $\nu(\text{N-H})$ of PAAm, suggesting that the AAm moieties consistently exist in the form of hydrogen bonds within the ionogel and hardly interact with cations/anions (Supplementary Fig. 5 and 6). This is also consistent with the poor miscibility between PAAm and ionic liquid. Therefore, based on the above analysis, we speculate that a 3D supramolecular network is formed between P(AA-co-AAm) and the ionic liquid (Fig. 1a). The rheological master curve reveals that the ionogel exhibits good thermal stability and remains in the rubbery state over a wide range of frequencies and temperatures (Supplementary Fig. 7a). By fitting the relationship between the shift factor and temperature (fitting temperature $> T_g + 50$ °C) using the Arrhenius equation, apparent activation energy of 59.3 kJ mol^{-1} is obtained (Supplementary Fig. 7b). This indicates that the supramolecular network present in the ionogel possesses dynamic properties, which can dissipate energy through the dissociation of the supramolecular interactions during deformation.”

We supplement the discussion on Page 14 of the revised manuscript.

“The X-ray photoelectron spectroscopy (XPS) results show that there is a tight bond between the ionogel and CF fabric. As shown in Supplementary Fig. 31, the C 1s peak of O-C=O shifts from 288.8 eV in CF and 288.5 eV in ionogel to 289.1 eV in FRIG, while the C 1s peak of C-OH shifts from 285.7 eV in CF to 285.4 eV in FRIG. Meanwhile, the O 1s peak of C=O in FRIG shifts to low binding energy while the O 1s peak of C-OH in FRIG shifts to high binding energy (Fig. 3b). The changes in C 1s peak and O 1s peak indicate a change in the electron density of the carboxyl and hydroxy groups due to the formation of strong binding between the polymer chains and CF. In addition, the cations and anions of the ionic liquid will further strengthen the bond between the ionogel and CF fabric. As illustrated in Supplementary Fig. 32, the

N 1s peak of the imidazole ring as well as the S 2p and F 1s peaks of TFSI⁻ in FRIG shifts to higher binding energy compared to those in the ionogel, indicating the ionic interaction between the ionogel and CF. Thus, a 3D supramolecular network is initially formed between the ionic liquid and polymer chains, followed by the creation of a tight interface through the synergistic interactions among the polymer chains, ionic liquid, and CF fabric, collectively forming the FRIG. CT results also demonstrate that the ionogel within the FRIG is tightly bound to the carbon fibers, with no gaps present (Supplementary Fig. 33).”

We supplement the discussion on Page 9 of the revised manuscript.

“Subsequently, we investigated the energy dissipation of FRIG during deformation through cyclic stress-strain curves (Supplementary Fig. 17a). Due to the tight bonding between the ionogel and CF fabric, FRIG can dissipate a significant amount of energy during deformation, with the dissipated energy increasing as the strain increases. Across various strains, the damping capacity (the ratio of dissipated energy to input energy) can surpass 80% (Supplementary Fig. 17b). The high dissipated energy and damping capacity contribute to the resistance of FRIG to external impacts and tearing.”

Question 2: The fundamental mechanism for how carbon fiber reinforces ionogel via charge interaction is not well demystify, where the authors only conduct a series of macro-level mechanical performance. That is, the manuscript lacks dedicated in-depth analysis which is necessary for publication in a journal like Nat. Commun.

Answer 2: Thanks for the Reviewer’s suggestion. We have supplemented experiments and elaborated on the structure of FRIG as well as the mechanism of its tear resistance.

1. Cross-linking structure between the ionogel and CF fabric: We have re-analyzed the XPS results and further discovered that CF fabric can simultaneously bind to the polymer chains and ionic liquids of the ionogel through hydrogen bonding and ionic interactions, resulting in extremely strong interfacial interlocking (Fig. 3b and Supplementary Fig. 31 and 32). Molecular dynamics simulations also indicate that there is a higher binding energy between CF fabric and ionogel in FRIG (Fig. 3c). Therefore, this leads to a tight and seamless bonding between the ionogel and CF fabric in FRIG, as confirmed by CT (Supplementary Fig. 33). Consequently, we have gained a clear

understanding of the micro-to-macro structure of FRIG: a 3D supramolecular network is formed through cation-oxygen interactions and hydrogen bonding between the ionic liquids and polymer chains within the ionogel; simultaneously, CF fabric is tightly bound to the ionogel through the synergistic interactions among polymer chains, ionic liquids, and CF fabric, collectively constituting FRIG.

Figure 3b. XPS O 1s spectra of the FRIG, ionogel, and CF fabric.

Supplementary Figure 31. XPS C 1s spectra of the FRIG, ionogel, and CF fabric.

Supplementary Figure 32. XPS N 1s (a), S 2p (b), F 1s (c) spectra of the FRIG, ionogel, and CF fabric.

Supplementary Figure 33. CT results of the FRIG. Scale bar: 500 μm .

2. High structural stability of the FRIG: Our previous results reveal that the tight bonding between CF fabric and the ionogel in FRIG exhibits high stability. During deformations induced by shearing, energy can be dissipated through bond breaking at the interface, leading to energy loss (Supplementary Fig. 35). Consequently, compared to the ionogel, FRIG exhibits a slight increase in storage modulus but a significant rise in loss modulus (at higher frequencies, the oscillations become more intense, resulting in more bond breaking and thus more energy dissipation). Therefore, the loss factor of FRIG increases significantly with frequency, enhancing its energy dissipation capacity, which further confirms the tight interfacial bonding between the ionogel and CF fabric. Moreover, the high modulus of CF and the strong bonding between CF fabric and the ionogel can also bolster the structural stability of FRIG. Hence, FRIG is resistant to stress relaxation and can maintain a high-stress level consistently (Supplementary Fig. 37).

3. Important role of the ionogel: Ionogels also play a crucial role in enhancing the tearing resistance of FRIG by effectively dissipating energy through the dissociation of supramolecular interactions during deformation, thereby increasing tearing toughness. We investigated the changes in internal interactions within the ionogel during deformation using infrared spectroscopy. As shown in Supplementary Fig. 38, the infrared results indicate that during deformation, the ionogel undergoes redshifts in the carboxyl $\nu(\text{C}=\text{O}/\text{PAA})$ as well as blueshifts in $\nu(\text{N}-\text{H})$, $\nu(\text{C}-\text{F})$ and $\nu(\text{S}=\text{O})$ due to the dissociation of cation-oxygen interactions and hydrogen bonds. Consequently, the energy dissipation of the ionogel during loading and unloading increases almost exponentially with strain, effectively dissipating energy at different stages of deformation (Fig. 3d and Supplementary Fig. 39). From a strain of 50% to 800%, the cyclic curves of the ionogel exhibit significant hysteresis, with the dissipated energy

increasing from 0.28 MJ m^{-3} to 19.2 MJ m^{-3} . Furthermore, at various strains, the damping capacity remains high ($> 85\%$), indicating that most of the input energy is used to dissociate supramolecular interactions for energy dissipation. Therefore, during deformation, the ionogel can also blunt crack propagation and inhibit the further spread of defects, exhibiting high tearing toughness (Supplementary Fig. 40). Additionally, the tight bonding between the ionogel and CF fabric prevents CF from being easily pulled out during deformation and can disperse the stress at the crack, reducing stress concentration.

Supplementary Figure 38. Infrared results of the ionogel at 0% and 500% strain. (a) ν (C=O/PAA); (b) ν (N-H); (c) ν (C-F) and ν (S=O).

Figure 3d. Dissipated energy and damping capacity of the ionogel at different strains.

Supplementary Figure 39. Cyclic stress-strain curves of the ionogel at different strains.

4. Mechanism analysis: Based on the above analysis, the mechanism underlying the excellent tearing resistance of FRIG is described as follows. The CF fabric can achieve strong interfacial bonding with the polymer chains and ions of the ionogel through synergistic effects. During the tearing process of FRIG, the tearing force initially causes significant deformation of the ionogel and distributes the force to a large energy-

dissipating area at the crack tip. During tearing, the ionogel effectively dissipates energy through the dissociation of supramolecular interactions, and the dissipated energy increases rapidly with increasing deformation. When the tearing force exceeds a threshold value, the ionogel at the crack tip begins to rupture. Therefore, the tearing toughness of FRIG is positively correlated with the tearing toughness of the ionogel. Throughout this process, the tight bonding between the CF fabric and the ionogel prevents interfacial separation and avoids the extraction of CF bundles. The tearing force, dispersed across the entire energy-dissipating area, is transferred to the CF bundles, causing them to fracture and releasing the large amount of elastic energy stored in the CF, which resists crack propagation. Therefore, the synergy between CF fabric and the ionogel significantly enhances the tearing resistance of FRIG.

We supplement the discussion on Page 14 of the revised manuscript.

“The X-ray photoelectron spectroscopy (XPS) results show that there is a tight bond between the ionogel and CF fabric. As shown in Supplementary Fig. 31, the C 1s peak of O-C=O shifts from 288.8 eV in CF and 288.5 eV in ionogel to 289.1 eV in FRIG, while the C 1s peak of C-OH shifts from 285.7 eV in CF to 285.4 eV in FRIG. Meanwhile, the O 1s peak of C=O in FRIG shifts to low binding energy while the O 1s peak of C-OH in FRIG shifts to high binding energy (Fig. 3b). The changes in C 1s peak and O 1s peak indicate a change in the electron density of the carboxyl and hydroxy groups due to the formation of strong binding between the polymer chains and CF. In addition, the cations and anions of the ionic liquid will further strengthen the bond between the ionogel and CF fabric. As illustrated in Supplementary Fig. 32, the N 1s peak of the imidazole ring as well as the S 2p and F 1s peaks of TFSI⁻ in FRIG shifts to higher binding energy compared to those in the ionogel, indicating the ionic interaction between the ionogel and CF. Thus, a 3D supramolecular network is initially formed between the ionic liquid and polymer chains, followed by the creation of a tight interface through the synergistic interactions among the polymer chains, ionic liquid, and CF fabric, collectively forming the FRIG. CT results also demonstrate that the ionogel within the FRIG is tightly bound to the carbon fibers, with no gaps present (Supplementary Fig. 33).”

We supplement the discussion on Page 15 of the revised manuscript.

“Besides the interaction between the ionogel and CF fabric, the ionogel also plays a

significant role in enhancing the tearing resistance of FRIG by effectively dissipating energy through the dissociation of supramolecular interactions during deformation. We investigated the changes in internal interactions within the ionogel during deformation using infrared spectroscopy. The infrared results indicate that during deformation, the ionogel undergoes redshifts in the carboxyl $\nu(\text{C=O/PAA})$ as well as blueshifts in $\nu(\text{N-H})$, $\nu(\text{C-F})$ and $\nu(\text{S=O})$ due to the dissociation of cation-oxygen interactions and hydrogen bonds (Supplementary Fig. 38). Consequently, the energy dissipation of the ionogel during loading and unloading increases almost exponentially with strain, effectively dissipating energy during deformation (Fig. 3d and Supplementary Fig. 39). From a strain of 50% to 800%, the cyclic curves of the ionogel exhibit significant hysteresis, with the dissipated energy increasing from 0.28 MJ m^{-3} to 19.2 MJ m^{-3} . Furthermore, at various strains, the damping capacity remains high ($> 85\%$), indicating that most of the input energy is used to dissociate supramolecular interactions for energy dissipation. Therefore, during deformation, the ionogel can also blunt crack propagation and inhibit the further spread of defects, exhibiting high tearing toughness (Supplementary Fig. 40).”

We supplement the discussion on Page 16 of the revised manuscript.

“Based on the above analysis, the mechanism underlying the excellent tearing resistance of FRIG is described as follows. The CF fabric can achieve strong interfacial bonding with the polymer chains and ions of the ionogel through synergistic effects. During the tearing process of FRIG, the tearing force initially causes significant deformation of the ionogel and distributes the force to a large energy-dissipating area at the crack tip. During tearing, the ionogel effectively dissipates energy through the dissociation of supramolecular interactions, and the dissipated energy increases rapidly with increasing deformation. When the tearing force exceeds a threshold value, the ionogel at the crack tip begins to rupture. Therefore, the tearing toughness of FRIG is positively correlated with the tearing toughness of the ionogel. Throughout this process, the tight bonding between the CF fabric and the ionogel prevents interfacial separation and avoids the extraction of CF bundles. The tearing force, dispersed across the entire energy-dissipating area, is transferred to the CF bundles, causing them to fracture and releasing the large amount of elastic energy stored in the CF, which resists crack propagation. Therefore, the synergy between CF fabric and the ionogel significantly enhances the tearing resistance of FRIG.”

Question 3: For the application section in Fig.4b, it obviously belongs to long-term cyclic test, so the anti-fatigue related properties must be considered in the previous mechanics section (Fig.2, 3).

Answer 3: Thanks for the Reviewer's suggestion. We tested the mechanical behavior and changes in energy dissipation capacity of FRIG over 5000 cycles using the tensile mode of DMA. As shown in Supplementary Fig. 19, during the cycling process, the modulus of FRIG remains stable. Meanwhile, the loss factor, which represents the energy dissipation capacity, also remains stable. This indicates that FRIG exhibits excellent fatigue resistance during cyclic tensile testing. Furthermore, the stability of the loss factor suggests that the interfacial bonding and energy dissipation capacity of FRIG remain stable during cyclic deformation.

Supplementary Figure 19. Modulus and $\tan \delta$ changes of the FRIG during cyclic stretching obtained by DMA.

We supplement the discussion on Page 9 of the revised manuscript.

“The changes in mechanical behavior and energy dissipation capacity of the FRIG after 5000 cycles were tested using the tensile mode of DMA. During the cycling process, both the modulus and loss factor of the FRIG remains stable, indicating its excellent fatigue resistance (Supplementary Fig. 19).”

Question 4: Demonstration in Supplementary Figure 33 and sensing section in Fig.4 is not convincing. Are the claimed mechanical performance combinations (strength, modulus, and toughness) indispensable for such kind of application? The previous 3 figures largely deal with mechanics, why it changes to electrical performance? Without

these rationales, these results are just a stacking of data and images.

Answer 4: Thanks for the Reviewer's suggestion. As we elaborated in the Introduction, ion-conductive materials have found wide applications in cutting-edge fields such as soft robotics, wearable devices, and flexible sensors due to their excellent mechanical and electrical properties. In these advanced applications, both good mechanical and electrical properties are indispensable, with mechanical properties providing physical support and electrical properties enabling sensing capabilities. Although numerous ion-conductive materials have been developed for these applications, current research primarily focuses on flexible soft materials, which are suitable for areas requiring soft mechanical properties, such as ionic skin and brain-machine interfaces. However, for applications relying on mechanical robustness, such as the fixation and motion monitoring of joints in intelligent soft robots, existing flexible soft materials are less suitable. They struggle to bear heavy loads and resist crack propagation. Although some flexible soft materials with high toughness and stretchability have been developed for biomimetic ligaments (*Nat. Commun.* **2022**, *13*, 2279; *Adv. Mater.* **2023**, *35*, 2210624), their strength is far inferior to human ligaments. Moreover, after multiple cycles, the yielding of these flexible soft materials leads to softening, making it difficult to effectively fix joints and perform sensing and monitoring, as demonstrated in Fig. 4e. Therefore, our claimed mechanical performance combinations are necessary for our applications.

Additionally, we have removed the fireproof demonstration image from Supplementary Figure 33 to better focus the main theme of this paper on mechanical properties and intelligent sensing.

Question 5: The ideal of this work is much like a previous work ((*Adv. Mater.* 2024, *36*, 2406252)), where the carbon fiber was used to reinforce polyurethane, and the reported mechanical metrics do not truly outperform the existing one. Also, in that reference, the underlying tearing resistance mechanism has been well demystified. In this context, the current work does not bring more new insight over previous one.

Answer 5: Thanks for the Reviewer's question. We have carefully read the literature you mentioned (*Adv. Mater.* **2024**, *36*, 2406252), which reports the preparation of carbon fiber-reinforced polymers (CFRPs) with high tearing toughness by compositing polyurethane with carbon fibers. Polyurethane contains nanodomains formed by phase

separation, which form tight interfacial bonding with the carbon fibers through hydrogen bonding. During deformation, energy is dissipated through the deformation, orientation, and dissociation of these nanodomains, leading to the fracture of CF bundles and thus achieving high tearing toughness in the CFRPs. The most outstanding CFRP exhibits a tearing toughness of 2012 kJ m^{-2} . The results presented in this article represent a significant advancement in the field of composites and provide valuable insights and directions for the design of tear-resistant CFRPs.

However, compared to these results, we believe our revised article also features several key differences and advancements. We would like to further elaborate on these differences and advancements in terms of material design, performance, mechanism, and applications, and to explain the importance and innovation of our revised manuscript to the Reviewer.

(1) In terms of material design, we have developed a novel class of ultra-tough composite ionogel materials through the tight interfacial bonding between ionogels with a 3D supramolecular network structure and fiber fabrics. Moreover, our strategy is not limited to carbon fibers and ionogels with a particular structure. Compared to the literature (*Adv. Mater.* **2024**, *36*, 2406252), our advancement lies in the versatility of the FRIG design, which can be applied to different types of fiber fabrics (such as carbon fibers and aramid fibers) as well as supramolecular ionogels with different types and chemical structures.

(2) In terms of performance, our FRIG exhibits remarkable tearing toughness, with a maximum tearing energy of the aramid fiber-reinforced one reaching 4219 kJ m^{-2} . Compared to the high tearing energy of 2012 kJ m^{-2} demonstrated by the CFRP in the literature (*Adv. Mater.* **2024**, *36*, 2406252), our performance is 2.1 times higher. Therefore, compared to currently reported fiber-reinforced materials, FRIG represents a significant advancement.

(3) In terms of the toughening mechanism, the abundant supramolecular interactions within the ionogel effectively dissipate energy through dissociation during deformation, while the tight interfacial bonding between the ionogel and CF ensures that they do not separate during deformation, thereby achieving uniform stress distribution. By dissipating the bulk energy of the ionogel and the elastic energy stored in the fabric, the tearing toughness of FRIG can be greatly enhanced. This differs significantly from the mechanism reported in the CFRP study (*Adv. Mater.* **2024**, *36*, 2406252), which primarily relies on the deformation, orientation, and dissociation of

microphase-separated nanodomains to dissipate energy. In contrast, our FRIG primarily relies on the dissociation of supramolecular interactions for energy dissipation. Although both mechanisms lead to similar results, namely the fracture of fiber bundles, they differ significantly at the microscopic level. Additionally, our FRIG represents a first universal example of toughening ionogels with high-performance fiber fabrics, providing new insights into the design of toughened flexible conductive gel materials. It also offers a valuable reference for high-performance composites and fiber-reinforced polymers.

(4) In terms of application advancements, due to its excellent mechanical properties, we have demonstrated that FRIG can be used as a bionic ligament for artificial bone fixation. Moreover, its superior electrical performance allows it to monitor movements in real time through electrical signals. Compared to conventional ion-conductive elastomer materials that tend to undergo yield softening after multiple deformations, FRIG exhibits higher stability and durability. Furthermore, compared to conventional ion-conductive materials, the high sensitivity of FRIG makes it easier to detect minute changes, laying the foundation for the application of FRIG in smart artificial ligaments and flexible sensors.

Therefore, from the perspectives of material design, performance, mechanism, and application, our FRIG is innovative and significant.

However, the literature (*Adv. Mater.* **2024**, *36*, 2406252) also represents a significant advancement in the field of composites. During the first review process of our article, we also simultaneously noticed the publication of this literature, and we are grateful to the Reviewer for highlighting its importance. Therefore, we have cited this literature in the introduction as a research development in the relevant field.

Response to the Reviewer #2

Comment: In this manuscript, Lyu and co-workers have reported a tough fiber-reinforced ionogels by incorporating high-performance fibers into elastic ionogels to efficiently dissipate energy. It exhibited extraordinary tearing toughness (4219 kJ m^{-2}), remarkable strength (365 MPa), high elastic modulus (1.0 GPa), and low bending modulus (2.2 MPa). Due to good ionic conductivity, fast response time, and exceptional strain sensitivity, the FRIG demonstrated great potential in intelligent robots and wearable devices. The following issues should be addressed before its acceptance.

Reply: We are very grateful to the Reviewer for the encouraging comments on our manuscript. We have revised the manuscript and hope to address the issues raised by the Reviewer.

Question 1: Some similar works have been previously described in another article published in Advanced Material (please refer to Adv. Mater. 2020, 32, 1907180; Adv. Mater. 2024, 36, 2406252). In this case, the authors have to further claim their advances compared with these works in Introduction.

Answer 1: Thanks for the Reviewer's suggestion. The advances relative to the existing literature have been added in the Introduction.

We supplement the sentence on Page 3 of the revised manuscript.

“In addition, compared to the current state-of-the-art fiber-reinforced elastomers (2500 kJ m^{-2}) or fiber-reinforced polyurethanes (2012 kJ m^{-2}),^{27,28} FRIG also exhibits a further improvement in tearing toughness (1.7–2.1 times).”

Question 2: The collagen fiber bundle in the tendon is anisotropic, whereas the carbon fiber in this paper is isotropic. The strengthening principles of these two materials are not compatible, so the authors may need to redefine them.

Answer 2: We are grateful for the suggestion regarding the distinction between our material design and the tendon structure. Based on the suggestion, we have removed the relevant statements about tendons in the Introduction. Additionally, we have modified the schematic diagram of the material design in Fig. 1, focusing primarily on the structural composition and internal supramolecular interactions of FRIG.

Figure 1a. Schematic of the FRIG composed of the CF fabric and supramolecular ionogel. The supramolecular ionogel is composed of the polymer chains and ionic liquid through the cation-oxygen interactions and hydrogen bonding.

Question 3: The thickness and fiber content of the composite ionogel should be provided.

Answer 3: Thanks for the Reviewer’s suggestion. The thickness of FRIG is 1 mm while the fiber fabric content is 33.9% (Supplementary Fig. 2).

Supplementary Figure 2. CF mass ratio in the FRIG.

We supplement the sentence on Page 5 of the revised manuscript.

“The thickness of the FRIG is controlled to 1 mm. The mass ratio of the CF fabric in the FRIG is calculated to be 33.9% (Supplementary Fig. 2).”

Question 4: What is the wettability of carbon fiber and premixed solution (the blend of

ionic liquids and monomers)?

Answer 4: Thanks for the Reviewer’s suggestion. The precursor solution exhibits good wettability with CF fabric. Photographic demonstrations and contact angle tests indicate that once the precursor solution comes into contact with CF fabric, it rapidly spreads out and can penetrate the carbon fibers (Supplementary Fig. 12).

Supplementary Figure 12. Photographs (a) and contact angle (b) of the precursor solution spreading on the CF fabric within 4 seconds

We supplement the sentence on Page 6 of the revised manuscript.

“As shown in Supplementary Fig. 12, the precursor solution exhibits excellent wettability with CF fabric, allowing it to rapidly penetrate into the interior of the fabric upon contact.”

Question 5: The authors need to provide CT images to observe the internal structure of the FRIG.

Answer 5: Thanks for the Reviewer’s suggestion. The CT results show that the CF fabric is present in the middle region of the composite, tightly bound to the ionogel without any voids (Supplementary Fig. 33). Cross-sectional images also reveal that the carbon fibers within the CF bundle are uniformly dispersed.

Supplementary Figure 33. CT results of the FRIG. Scale bar: 500 μm.

We supplement the sentence on Page 14 of the revised manuscript.

“CT results also demonstrate that the ionogel within the FRIG is tightly bound to the carbon fibers, with no gaps present (Supplementary Fig. 33).”

Question 6: In Figure S14, there is not a significant transition from being pulled to fracture of the carbon fiber from 30 to 70 mm. The authors should show this transition in a more intuitive way.

Answer 6: Thanks for the Reviewer’s suggestion. We have retaken photographs of FRIG with different widths during the tearing process. As shown in Supplementary Fig. 22, when the width ranges from 10 mm to 30 mm, the CF bundle is mainly pulled out, and its morphology remains largely intact. At a width of 40 mm, both fracture and pull-out of the CF bundle coexist. As marked by the red line, part of the CF bundle is broken, while another part is pulled out. When the width reaches 50 mm, the main portion where the CF bundles are in contact with each other is basically completely torn, with no CF bundle left being pulled out. As indicated by the red line, the CF bundle breaks earlier closer to the location of the notch, whereas it breaks later farther away from the notch.

Supplementary Figure 22. Photographs of FRIG-0.85-60% with different sample widths after trouser tearing. With increasing the sample width, the tearing behavior transforms from the fiber pullout to the fiber fracture.

Question 7: The strongly adherent between carbon fiber and ionic gel has an important effect on the mechanical properties of FRIG. It is necessary to provide a interlayer shear test for more detailed verification, and the corresponding standard test method should be described in detail.

Answer 7: Thanks for the Reviewer’s suggestion. We further investigated the strong adhesion between the CF fabric and the ionogel through lap shear experiments. The adhesion strength was then calculated and found to be 1.2 MPa (Supplementary Fig.

15).

Supplementary Figure 15. Adhesion between the CF fabric and the ionogel through lap-shear test.

We supplement the experimental details on Page 22 of the revised manuscript.

“The ionogel precursor solution was brushed between two pieces of CF fabric to form a bonded area of $1 \times 1 \text{ cm}^2$, followed by in-situ polymerization. After the polymerization was completed, a universal testing machine was used to conduct tensile testing under ambient conditions at a speed of 50 mm min^{-1} . The adhesion strength was calculated through the force divided by the contact area.”

We supplement the discussion on Page 7 of the revised manuscript.

“In addition, the lap-shear test indicates that there remains good adhesion between the CF fabric and the ionogel when the CF bundle is replaced with CF fabric (Supplementary Fig. 15).”

Question 8: In Figure 2e, the authors should provide an explanation on the maximum tearing toughness while $x = 60\%$, $f = 0.85$.

Answer 8: Thanks for the Reviewer’s suggestion. CF fabric can achieve strong interfacial bonding with ionogels. During the tearing process of FRIG, the tearing force first causes significant deformation of the ionogel and effectively dissipates energy through the dissociation of supramolecular interactions. In this process, the tight bonding between the CF fabric and the ionogel causes the CF to break rather than be pulled out, thereby releasing the elastic energy stored in the CF. Therefore, when the CF bundle is in fracture mode, the tearing toughness of FRIG positively correlates with the tearing toughness of the ionogel, as demonstrated by the relationship shown in

Figure 3a. Consequently, since the ionogel with $x = 60\%$ and $f = 0.85$ exhibits the highest tearing toughness, the corresponding FRIG also possesses the highest tearing toughness.

We supplement the sentence on Page 13 of the revised manuscript.

“Therefore, FRIG-0.85-60% has the highest tearing toughness due to the highest tearing toughness of IG-0.85-60% among all ionogels.”

Question 9: In Figure 2e and S10, why does the proportion of polymer monomers have little effect on tensile strength and elongation at break, while having a greater effect on trouser tearing?

Answer 9: Thanks for the Reviewer’s suggestion. CF fabric can achieve strong interfacial bonding with ionogels. During the tearing process of FRIG, the tearing force effectively dissipates energy through significant deformation of the ionogel and fracture of the CF bundle. Therefore, the toughness and energy dissipation process of the ionogel have a significant impact on the tearing toughness of FRIG. Then, the proportion of polymer monomers has a great effect on trouser tearing.

However, for tensile strength and elongation at break, their performance mainly depends on the bridging effect of the ionogel between CF bundles. During tensile failure, the ionogel mainly transfers the force of the broken fibers to adjacent fibers through relatively small shear deformation. In other words, the load-bearing capacity lost due to fiber fracture is mainly compensated by fibers in adjacent areas, thereby delaying material failure, and enhancing strength. The primary role of the ionogel is to bridge different CF bundles. The degree of deformation of the ionogel during tensile failure is much less than the significant deformation during tearing. Therefore, the energy dissipation of the ionogel during tensile failure is also relatively small, and energy dissipation is mainly generated by the fracture of carbon fibers. The modulus differences among ionogels with different monomer ratios are relatively small, which may result in insignificant variations in their bridging effects. Conversely, for ionogels with varying ionic liquid contents, the substantial modulus differences may lead to more pronounced differences in their bridging effects. Consequently, the difference in monomer ratio has a minor impact on tensile strength and elongation at break.

Question 10: In Figure 3b, the results of XPS only prove that carbon fibers interact with PAA and PAAm. The authors should provide evidence to prove the interaction between ionic liquid and carbon fiber.

Answer 10: Thanks for the Reviewer's suggestion. We have re-analyzed the XPS results and further discovered that carbon fibers can simultaneously bond with both the polymer chains and ionic liquids of the ionogel, resulting in extremely strong interfacial interlocking (Fig. 3b, Supplementary Fig. 31 and 32). In other words, a tight bond is formed between CF fabric and the ionogel through the synergistic interactions among polymer chains, ionic liquids, and CF fabric, collectively constituting FRIG.

Figure 3b. XPS O 1s spectra of the FRIG, ionogel, and CF fabric.

Supplementary Figure 31. XPS C 1s spectra of the FRIG, ionogel, and CF fabric.

Supplementary Figure 32. XPS N 1s (a), S 2p (b), F 1s (c) spectra of the FRIG, ionogel, and CF fabric.

We supplement the discussion on Page 14 of the revised manuscript.

“The X-ray photoelectron spectroscopy (XPS) results show that there is a tight bond between the ionogel and CF fabric. As shown in Supplementary Fig. 31, the C 1s peak of O=C=O shifts from 288.8 eV in CF and 288.5 eV in ionogel to 289.1 eV in FRIG, while the C 1s peak of C-OH shifts from 285.7 eV in CF to 285.4 eV in FRIG. Meanwhile, the O 1s peak of C=O in FRIG shifts to low binding energy while the O 1s peak of C-OH in FRIG shifts to high binding energy (Fig. 3b). The changes in C 1s peak and O 1s peak indicate a change in the electron density of the carboxyl and hydroxy groups due to the formation of strong binding between the polymer chains and CF. In addition, the cations and anions of the ionic liquid will further strengthen the bond between the ionogel and CF fabric. As illustrated in Supplementary Fig. 32, the N 1s peak of the imidazole ring as well as the S 2p and F 1s peaks of TFSI⁻ in FRIG shifts to higher binding energy compared to those in the ionogel, indicating the ionic interaction between the ionogel and CF. Thus, a 3D supramolecular network is initially formed between the ionic liquid and polymer chains, followed by the creation of a tight interface through the synergistic interactions among the polymer chains, ionic liquid, and CF fabric, collectively forming the FRIG.”

Question 11: In Figure 3d, the relationship between damping capability and interfacial bonding should be further explained.

Answer 11: Thanks for the Reviewer’s suggestion. The tight bonding between CF fabric and the ionogel in FRIG exhibits high stability. When undergoing deformations through shearing, energy can be dissipated through bond breaking at the interface, leading to energy loss. Therefore, compared to the ionogel, FRIG exhibits a slight increase in storage modulus but a significant rise in loss modulus (at higher frequencies, the oscillations become more intense, resulting in more bond breaking and thus more energy dissipation) (Supplementary Fig. 35). Consequently, the loss factor of FRIG increases significantly with frequency, indicating enhanced energy dissipation capability, which further proves the tight interfacial bonding between the ionogel and CF fabric.

This result is further confirmed by the PDMS/CF comparative sample. Due to the insufficient interfacial bonding between PDMS and CF fabric, the increase in storage modulus and loss modulus of PDMS/CF is mainly attributed to the overall modulus enhancement of the composite (Supplementary Fig. 36). During shear deformation, the weak interfacial bonding between PDMS and CF dissipates less energy. Therefore, the loss modulus does not change significantly with frequency. Ultimately, the loss factor of PDMS/CF remains at a low level, similar to that of PDMS, which reflects the weak interfacial bonding between PDMS and CF.

We modify the discussion on Page 14 of the revised manuscript.

“Meanwhile, the rheological results demonstrate that the loss modulus of the elastic ionogel rises rapidly once it is complexed with CF due to the bond breaking at the interface during deformation (Supplementary Fig. 35). Moreover, the loss factor of FRIG increases significantly with frequency, indicating enhanced energy dissipation capability, which further proves the tight interfacial bonding between the ionogel and CF fabric.”

Question 12: In Figure 4, what is the conductivity and sensitivity of the FRIG after 10,000 bending cycles, the author should provide the corresponding experimental data.

Answer 12: Thanks for the Reviewer’s suggestion. We have supplemented the conductivity and sensitivity of FRIG after 10,000 bending cycles. As shown in Supplementary Fig. 46, FRIG can still maintain high conductivity and sensitivity, indicating that its conductive pathways remain uncompromised.

Supplementary Figure 46. (a) Room-temperature ionic conductivity of FRIG-0.85-60% after bending 10000 cycles. (b) GF values at different strains of FRIG-0.85-60% after bending 10000 cycles.

We supplement the discussion on Page 17 of the revised manuscript.

“In addition, FRIG can still maintain high conductivity and sensitivity after 10,000 bending cycles, indicating that its conductive pathways remain uncompromised (Supplementary Fig. 46).”

Question 13: After 10,000 cycles of testing under stretching conditions, whether the carbon fiber and ionogel are separated at the interface?

Answer 13: Thanks for the Reviewer’s suggestion. After 10,000 cycles, the bonding between the CF fabric and the ionogel remains tight (Supplementary Fig. 45). Photographs show that the morphology of FRIG remains intact. SEM results reveal no separation between the CF fabric and ionogel at the interface.

Supplementary Figure 45. Photograph (a) and SEM image (b) of the FRIG after bending 10,000 cycles.

We supplement the discussion on Page 17 of the revised manuscript.

“Moreover, the ionogel and CF fabric remain tightly bound together without separation (Supplementary Fig. 45).”

Question 14: There are some mistakes in this manuscript in line 66, line 117, and line 234, the authors should check the full text carefully.

Answer 14: Thanks to the Reviewer for pointing out the errors. We have checked the entire text to ensure that all written content and numbers are consistent.

Response to the Reviewer #3

Comment: The authors have developed tough fiber-reinforced ionogels with notable crack resistance by incorporating high-performance fibers into elastic ionogels. However, the novelty of this strategy needs to be highly improved.

Reply: We are very grateful to the Reviewer for the comments on our manuscript. We have revised the manuscript and hope to address the issues raised by the Reviewer.

Regarding the novelty of this strategy, we would like to further clarify to the Reviewer in terms of material design, performance, mechanism, and applications based on the revised manuscript.

(1) In terms of material design, we have developed a novel class of ultra-tough composite ionogel materials through the tight interfacial bonding between ionogels with a 3D supramolecular network structure and fiber fabrics. Moreover, our strategy is not limited to carbon fibers and ionogels with a particular structure. This strategy can be applied to different types of fiber fabrics (such as carbon fibers and aramid fibers) as well as supramolecular ionogels with different types and chemical structures.

(2) In terms of performance, our FRIG exhibits remarkable tearing toughness, with a maximum tearing energy of aramid fiber-reinforced one reaching 4219 kJ m^{-2} . Compared to currently reported fiber-reinforced materials, FRIG represents a significant advancement.

(3) In terms of the toughening mechanism, the abundant supramolecular interactions within the ionogel effectively dissipate energy through dissociation during deformation, while the tight interfacial bonding between the ionogel and CF ensures that they do not separate during deformation, thereby achieving uniform stress distribution. By dissipating the bulk energy of the ionogel and the elastic energy stored in the fabric, the tearing toughness of FRIG can be greatly enhanced. Our FRIG represents a first universal example of toughening ionogels with high-performance fiber fabrics, providing new insights into the design of toughened flexible conductive gel materials. It also offers a valuable reference for high-performance composites and fiber-reinforced polymers.

(4) In terms of application advancements, due to its excellent mechanical properties, we have demonstrated that FRIG can be used as a bionic ligament for artificial bone fixation. Moreover, its superior electrical performance allows it to monitor movements in real time through electrical signals. Compared to conventional ion-conductive elastomer materials that tend to undergo yield softening after multiple

deformations, FRIG exhibits higher stability and durability. Furthermore, compared to conventional ion-conductive materials, the high sensitivity of FRIG makes it easier to detect minute changes, laying the foundation for the application of FRIG in smart artificial ligaments and flexible sensors.

The above summarizes the innovation and importance of our revised manuscript. Once again, we are deeply grateful to the Reviewer for spending the valuable time to review our work and for providing invaluable suggestions.

Question 1: On page 2, lines 21-23, the authors claim that fiber-reinforced ionogels can efficiently dissipate energy. However, there is no supporting data in the manuscript to substantiate this conclusion, such as cyclic loading-unloading tests.

Answer 1: Thanks for the Reviewer’s suggestion. We conducted a study on energy dissipation using cyclic stress-strain curves (Supplementary Fig. 17). The results reveal that the FRIG can dissipate a significant amount of energy during deformation, with the dissipated energy increasing as the strain increases. Meanwhile, across different strains, the damping capacity (the ratio of dissipated energy to input energy) can reach 80%. High energy dissipation aids FRIG in effectively resisting external impacts and tearing.

Supplementary Figure 17. (a) Cyclic stress-strain curves of FRIG at different strains; (b) Corresponding dissipated energy and damping capacity at different strains.

We supplement the discussion on Page 9 of the revised manuscript.

“Subsequently, we investigated the energy dissipation of FRIG during deformation through cyclic stress-strain curves (Supplementary Fig. 17a). Due to the tight bonding between the ionogel and CF fabric, FRIG can dissipate a significant amount of energy during deformation, with the dissipated energy increasing as the strain increases. Across various strains, the damping capacity (the ratio of dissipated energy to input energy)

can surpass 80% (Supplementary Fig. 17b). The high dissipated energy and damping capacity contribute to the resistance of FRIG to external impacts and tearing.”

Question 2: Several images in the manuscript lack a scale bar, including Figure 1d, Figure 1e, Figure 4d, and Supplementary Figure 2, which compromises the clarity and reproducibility of the presented data.

Answer 2: Thanks for the Reviewer’s suggestion. The scale bar has been added to the relevant images.

Figure 1. (b) Photographs of the FRIG showing the bending and distorting processes. (c) Photographs of the FRIG showing extraordinary crack resistance.

Figure 4. (d-e) Photographs of the FRIG (d) and ionogel (e) fixing artificial bones.

Supplementary Figure 3. Photographs of ionogels with different AA ratios. (a) Ionogel composed of PAA and [EMIM][OTf]. (b) Ionogel composed of PAAm and [EMIM][OTf]. (c) Ionogels composed of P(AA-co-AAm) and [EMIM][OTf] with different AA ratios and 60% IL content. The copolymer has good compatibility with the ionic liquid.

Question 3: On page 7, lines 119-136, the manuscript states that the strength of CF fabric (184 MPa) is greater than the interfacial bonding strength (4.8 MPa) and the strength of the ionogel (2.8 MPa). This contradicts the assertion that "the fiber bundles break rather than being pulled out."

Answer 3: Thanks for the Reviewer’s question. We apologize for any confusion caused

by our previous unclear expressions. The 4.8 MPa refers to the interfacial adhesion strength between the CF bundle and the ionogel when they are in contact. When the CF bundle is bonded with ionogel and undergoes deformation, the interfacial adhesion strength (4.8 MPa) is higher than the fracture strength of the ionogel (2.8 MPa). Therefore, during deformation, interfacial debonding between the ionogel and CF bundle does not occur; instead, significant deformation of the ionogel occurs to dissipate energy.

On the other hand, 184 MPa represents the tensile strength of the CF fabric during tensile failure, which is calculated using the cross-sectional area of the fracture surface. However, when the CF bundle is combined with the ionogel, the contact area between the ionogel and CF bundles is much larger than their cross-sectional area. As shown in Fig. 2a and Supplementary Fig. 14, as long as the embedded depth of the CF bundle in the ionogel exceeds 5 mm, the CF bundle will fracture when being pulled out of the ionogel. Since all our FRIG sample strips are larger than 5 mm (with tearing sample widths of 50 mm and tensile sample widths of 10 mm), this ensures that our sample primarily undergoes fracture rather than pullout during deformation.

We supplement the discussion on Page 7 of the revised manuscript.

“Therefore, during the deformation process, interfacial detachment between the ionogel and CF does not occur; instead, intense deformation of the ionogel primarily occurs to dissipate energy. Meanwhile, the strong adhesion between the ionogel and the CF bundle ensures that the adhesive force between them is greater than the breaking force of the CF bundle when stretched.”

Question 4: On page 8, lines 136-140, it is mentioned that the strength of FRIG-0.85-60% increased to 315 MPa. A more detailed explanation of the mechanism behind the fiber-reinforced ionogel would be beneficial, as the strength of the ionogel may predominantly derive from the carbon fiber.

Answer 4: Thanks for the Reviewer’s suggestion. The improvement of the tensile strength is mainly caused by the bridging effect between the ionogel and CF fabric.

We supplement the discussion on Page 9 of the revised manuscript.

“The improvement of the mechanical properties of FRIGs originates from the bridging

effect of the ionogel between the CF bundles.²⁹ During the fracture process, the ionogel can transfer the force from the broken fibers to the adjacent fibers through shear deformation, *i.e.*, the lost loading capacity due to fiber breakage can be made up by the fibers in the adjacent region. Thus, fiber breakage is not concentrated on a limited scale, and a larger overload region can be generated to delay material failure and increase the strength.”

Question 5: The manuscript lacks information on the mass ratio of carbon fiber to ionogel. Including this data would provide a better understanding of the primary factors contributing to the high strength of the material.

Answer 5: Thanks for the Reviewer’s suggestion. The fiber fabric content in the FRIG is 33.9% (Supplementary Fig. 2).

Supplementary Figure 2. CF mass ratio in the FRIG.

Response to Reviewers

We are appreciative of the feedback from our peer reviewers. Their constructive comments have significantly contributed to enhancing our manuscript. In response to their concerns, we have expanded our research with additional experiments, analyses, and discussions. These improvements are thoroughly integrated into the revised manuscript. In addition, based on Reviewer #1's suggestion, the abbreviation FRIG (Fiber-reinforced ionogel) in the response letter and revised manuscript has been changed to FRCI (Fiber-reinforced composite ionogel) to better reflect the composite nature of the fiber fabric and ionogel.

To facilitate a clear understanding of our revisions, we have organized our responses as follows:

Reviewer Comments and Our Responses: Each comment from the Reviewers (presented in black text) is followed by our corresponding response (in blue text).

Highlighted Text in the Revised Manuscript: Text modifications in the revised manuscript are red in the response letter.

Response to the Reviewer #1

Reply: We are deeply grateful to the Reviewer for the suggestions on our manuscript. We have revised the manuscript according to the Reviewer's suggestion and hope to address all concerns of the Reviewer.

Question 1: For finite-element modeling, there are some notable yellow spots in Fig.3e, corresponding to the higher stress concentration, whereas there is overall blue one in the PDMS/CF (3f), this analysis is contrast to the experimental result (the stress at the crack tip in the FRIG is much more dispersed than that in PDMS/CF.). Besides, there are some typos (page 16, there are no fig. 3 g and 3h).

Answer 1: Thanks for the Reviewer's suggestion. Due to the higher stress concentration in PDMS/CF, the Mises stress scales from the finite element simulation results differ between FRCI and PDMS/CF. Specifically, the stress scale for FRCI ranges from 0 to 0.3, whereas for PDMS/CF, it ranges from 0 to 3.0 (as shown in the right parts of Fig. 3e and 3f). Although there are two yellow spots in FRCI, the corresponding stresses are

still far less than those represented by the green spots in PDMS/CF. Additionally, both the yellow spots in FRCI and the green spots in PDMS/CF correspond to the crack tips during the tearing process and the locations where the material is in contact with the clamps. Therefore, it is consistent with the experimental result that the stress at the crack tip in the FRCI is much more dispersed than that in PDMS/CF.

Furthermore, thanks to the Reviewer for pointing out the typos. We have corrected them, with Figures 3g and 3h now appropriately labeled as 3e and 3f, respectively.

We modify the sentences on Page 16 of the revised manuscript.

“For the FRCI, stress can be transferred around the crack tip through the ionogel to reduce stress concentration (Fig. 3e).”

“However, for PDMS/CF, the maximum stress is concentrated at the crack tip (Fig. 3f).”

Question 2: ‘Moreover, the FRIG has a highly sensitive mechanical responsiveness and its resistance increases rapidly with the increase of strain, which can be used as a strain sensor to percept external stimuli and demonstrates its potential as a smart material.’ FRIG shows high crack resistance with large energy dissipation, could the authors please explain how to understand the term of ‘mechanical responsiveness’

Answer 2: Thanks for the Reviewer’s suggestion. Here, we intended to express that the resistance signal of FRCI exhibits sensitive changes with increasing strain, as similarly stated in some literature (*Nat. Commun.* **2023**, *14*, 219; *Adv. Funct. Mater.* **2022**, 2204366). However, after reviewing the literature, we found that the term “mechanical response” is also often used to describe the mechanical behavior of materials in many contexts. Therefore, to convey our meaning more accurately and avoid any misunderstanding by readers, we have changed this term to “electromechanical response.” (*Science* **2016**, *354*, 1257-1260)

We modify the sentence on Page 17 of the revised manuscript.

“Moreover, the FRCI has a highly sensitive electromechanical response and its resistance increases rapidly with the increase of strain, which can be used as a strain

sensor to percept external stimuli and demonstrates its potential as a smart material.”

Question 3: For strain sensor section, some preliminary biocompatibility/cell toxicity test should be considered if they develop its application as artificial bones. The authors examined the temperature dependent conductivity, so how about the sensing signal over the such temperature range. Besides, the authors highlight its excellent GF over previous references, while the superficial application in Fig.4f by only installing on a robot toy cannot reflect this metric. Moreover, just mentioned in initial comment, the strain scenarios in Fig.4f and Fig.S44 does not require the sensors possessing the high crack resistance. The authors must consider this point well.

Answer 3: Thanks for the Reviewer’s suggestion. We conducted tests on the cell toxicity of FRCI and the results indicate that it exhibits excellent biocompatibility (Supplementary Fig. 48). Furthermore, within the temperature range we tested, FRCI demonstrates good sensitivity over a wide temperature range (Supplementary Fig. 46).

Supplementary Figure 48. Cell viability incubated with different amounts of FRCI for 24 h.

Supplementary Figure 46. GF values of FRCI-0.85-60% at different temperatures.

Furthermore, as we have emphasized, FRCI is designed for applications requiring high strength and tear resistance, such as the bionic bone fixation illustrated in Fig. 4d, where it mimics the function of ligaments. Material failure can be prevented by possessing excellent mechanical properties during the sensing of vigorous activities (*Adv. Mater.* **2023**, 2304145). Otherwise, issues such as elastomer softening (Fig. 4e) or material fracture (Supplementary Fig. 53) may occur. Therefore, for FRCI, its primary function is to securely fix artificial bones, achieving a similar role to ligaments, with the secondary capability of strain sensing. Hence, the strain sensing scenarios presented in Fig. 4f and Supplementary Fig. 49 primarily focus on areas requiring high mechanical performance, such as elbows and knees. The strain sensing in these parts has high requirements for the mechanical properties of the materials (*Adv. Mater.* **2023**, 2304145). Conventional soft elastomers struggle to effectively fixate in these parts, leading to softening (Fig. 4e).

Moreover, according to the Reviewer's suggestion, We further investigated the ability of FRCI to fix artificial bone in the presence of cracks for verifying the effectiveness of crack resistance. In the presence of cracks, conventional elastomers are prone to failure due to crack propagation (Supplementary Fig. 54b). In contrast, FRCI remains functional (Supplementary Fig. 54a). Therefore, the crack resistance can prevent accidents, extend life, and improve reliability.

Supplementary Figure 54. Photographs of using FRCI (a) and ionogel (b) both with cracks as artificial ligaments to fix artificial bones.

Furthermore, in the field of actuators or sensing applications that require impact resistance, ionogels are equally in need of possessing good mechanical strength and tear resistance (*Nature* **2024**, *631*, 313–318; *Adv. Sci.* **2022**, 2207233). Therefore, in response to the Reviewer’s suggestions regarding applications, we carefully considered scenarios requiring high mechanical performance and high crack resistance, thereby expanding the application of FRCI in self-sensing impact protection. As shown in Supplementary Fig. 55a, when using FRCI as a protective material, it can effectively block external impacts. Moreover, it can spontaneously sense the timing and intensity of external impacts (Supplementary Fig. 55b). Simultaneously, even in the presence of cracks, FRCI retains its ability to resist external impacts, with its sensing capability remaining similar to that without cracks (Supplementary Fig. 55c). In contrast, conventional ionogel materials are prone to localized damage when subjected to high external impact forces (Supplementary Fig. 55d).

Supplementary Figure 55. Self-sensing impact protection application. (a) Schematic of the impact process. (b) Resistance signals of the impact using steel balls with different weights. (c) Resistance signals of the impact using steel balls with different weights in the presence of cracks. (d) Ionogel underwent rupture after being subjected to impact.

We add the sentence on Page 17 of the revised manuscript.

“Furthermore, the FRCI has good biocompatibility (Supplementary Fig. 48).”

“In addition, the sensing signal remains sensitive over a wide temperature range (Supplementary Fig. 46).”

“Additionally, even in the presence of cracks, FRCI can still safeguard artificial bones from fracturing, unlike traditional elastomers which are prone to breaking (Supplementary Fig. 54).”

We add the sentence on Page 18 of the revised manuscript.

“Meanwhile, regardless of whether cracks exist in FRCI, it can spontaneously perceive the timing and intensity of external impacts while providing impact protection, thereby enhancing reliability and extending service life (Supplementary Fig. 55a-c). In contrast,

traditional ionogels are prone to damage when subjected to excessive impacts (Supplementary Fig. 55d).”

Question 4: For the fatigue resistance, some basic parameters, such as fatigue threshold, must be considered. More specific parameters should be examined after long-term loading cycles.

Answer 4: Thanks for the Reviewer’s suggestion. We tested the fatigue resistance of FRCI using the single-notched samples at the strain of 5%, 10%, 15%, 18%, and 20% with a constant crack length. The crack propagation appears at the strain of 18%. Thus, the fatigue threshold can reach 26.4 kJ m^{-2} (Supplementary Fig. 28). This fatigue threshold surpasses that of many tear-resistant ionogels/elastomers (*Adv. Mater.* **2024**, *36*, 2309576; *Nat. Commun.* **2022**, *13*, 4411), further demonstrating the effectiveness of our strategy.

Supplementary Figure 28. Crack extension per cycle dc/dN versus applied energy release rate G for the FRCI.

We add the sentence on Page 11 of the revised manuscript.

“Meanwhile, FRCI also exhibits excellent fatigue resistance with a fatigue threshold as high as 26.4 kJ m^{-2} (Supplementary Fig. 28).”

Question 5: Considering such a high fiber content (34 wt%), it is doubtful to define it

as ionogels (elastomer/composite ionogel may be more suitable).

Answer 5: Thanks for the Reviewer’s suggestion. We have redefined this category of materials and named them “Fiber-reinforced composite ionogel (FRCI)”. This name not only reflects the fiber-reinforced characteristics but also highlights that the material is a composite composed of ionogels. Additionally, we have made corresponding modifications (from “FRIG” to “FRCI”) to the text descriptions and figures in both the manuscript and the supplementary information.

Question 6: The authors may also consider the temperature dependent mechanical performance over a wide range.

Answer 6: Thanks for the Reviewer’s suggestion. We have conducted further tests on temperature-dependent mechanical properties, including tensile curves and tearing curves at 50 °C and 80 °C.

Supplementary Figure 17. Stress-strain curves of FRCI-0.85-60% at 50 °C and 80 °C.

Supplementary Figure 26. Tearing curves (a) and tearing toughness (b) of FRCI-

0.85-60% at 50 °C and 80 °C.

We add the sentence on Pages 9 and 10 of the revised manuscript.

“Furthermore, even when the temperature rises to higher levels, such as 50 °C and 80 °C, FRCI still maintains good load-bearing capacity. Compared to room temperature, the increase in temperature results in a slight decrease in the modulus and a slight increase in the stretchability (Supplementary Fig. 17).”

“Additionally, temperature variations do not affect the tearing toughness of FRCI, with the toughness at 50 °C and 80 °C being almost identical to that at room temperature (Supplementary Fig. 26).”

Question 7: Page 2 ‘The inherent fragility of supramolecular bonds also poses a limitation on the strength and toughness’, in fact, there are abundant references about tough and strong elastomers based on supramolecular bonds.

Answer 7: Thanks for the Reviewer’s suggestion. What we mean here is that although the disruption of supramolecular interactions can dissipate energy to improve the strength and toughness of materials, it remains challenging to achieve the performance levels of some ultra-strong materials (such as the FRCI mentioned in this paper, as well as metals and engineering plastics) solely through supramolecular interactions. However, the Reviewer’s suggestion is well-taken, as the design of supramolecular interactions is indeed an effective approach in high-strength and high-toughness elastomers. Therefore, we have revised the relevant sentence to more accurately convey our meaning and avoid any misunderstanding by readers.

We modify the sentence on Page 2 of the revised manuscript.

“While supramolecular interactions can enhance the strength and toughness of materials by dissipating energy, the design of ultra-tough ICMs remains limited.”

Question 8: The stress value for FRIG does not agree with each other, where it reaches

to 120 MPa in loading-unloading curves (Fig.S17), whereas it is only about 90 MPa in Fig.2b.

Answer 8: Thanks for the Reviewer's suggestion. Mechanical property tests typically exhibit fluctuations within a certain average range. In our study, all mechanical property results were obtained by testing samples prepared from different batches and then calculating the statistical average, in order to minimize errors and randomness. However, when supplementing the energy dissipation results in Fig. S18 during the first round of revisions, we only used samples from one batch. Therefore, we are grateful for the Reviewer's valuable suggestion and have conducted further parallel experiments. The mechanical strength observed in the cyclic stress-strain curves is similar to the results presented in Fig. 2b. Additionally, we have re-presented the dissipated energy and damping capacity using averages with error bars.

Supplementary Figure 18. (a) Cyclic stress-strain curves of FRCI at different strains; (b) Corresponding dissipated energy and damping capacity at different strains.

Question 9: In the reply ‘Moreover, after multiple cycles, the yielding of these flexible soft materials leads to softening, making it difficult to effectively fix joints and perform sensing and monitoring, as demonstrated in Fig. 4e.’ The authors fail to prove the specific mechanics of the FRIG after cyclic loading. In addition, ‘our FRIG represents a first universal example of toughening ionogels with high-performance fiber fabrics, providing new insights into the design of toughened flexible conductive gel materials.’ It should be noted that the prototype of mechanical reinforcement for gel using fabric matrix can be traced by to Gong et al in 2016 (<https://doi.org/10.1002/adfm.201605350>, <https://doi.org/10.1039/C5MH00127G>)

Answer 9: Thanks for the Reviewer's suggestion. We tested the stress-strain curve of

FRCI after bending it 10,000 times and found that it can still maintain good mechanical properties (Supplementary Fig. 50).

Supplementary Figure 50. Stress-strain curve of FRCI-0.85-60% after bending 10000 times.

Furthermore, we appreciate the Reviewer's mention of Gong *et al.*'s work. Indeed, Gong *et al.*'s research on toughening hydrogels using fabrics provided us with some inspiration, and we have cited some of their earlier works in the manuscript (*e.g.*, *Adv. Funct. Mater.* **2017**, 27, 1605350; *J. Mater. Chem. A* **2019**, 7, 13431-13440). Gong *et al.*'s research primarily focuses on toughening hydrogels using fabrics. However, due to the weak interaction between hydrogels and fabrics, the composites are prone to fiber pullout during tearing, which limits the effective enhancement of toughness. Taking this into consideration and building on our research foundation in the field of ionogels, we toughened ionogels using fabrics, achieving effective interfacial interlocking through supramolecular interactions between them. This allows for fiber fracture rather than pullout during deformation, significantly improving the material's tear resistance. Moreover, the strategy of toughening ionogels using fabrics is universal. In addition, compared to hydrogels, the ions in ionogels can also serve as conductive media (although hydrogels can also achieve conductivity by adding ionic salts, Gong *et al.* found that the introduction of free ions can disrupt the electrostatic interactions between polymer chains in polyampholyte hydrogels with toughening functions, leading to decreased performance). Therefore, this was the main point we intended to convey in our previous reply, namely that we pioneered the toughening of ionogels using fabrics, achieving significantly stronger performance compared to hydrogels, and providing new insights into the design of toughened ionogel materials.

We add the sentence on Page 18 of the revised manuscript.

“The stress-strain curve reveals that the strength and modulus of the composite ionogel only exhibit a slight decrease after bending 10,000 times (Supplementary Fig. 50).”

Question 10: According to maximal strain ratio for FRIG is less than 20% (Fig.S16 a, b), whereas for the body motion strain sensor in Fig.4, normally the tolerant strain range for body joint must be large than 30%, this result indicates that the achieved FRIG may not suitable for body motion monitor.

Answer 10: Thanks for the Reviewer’s suggestion. After conducting an online search, we have found that the tensile range typically varies among different body parts. For instance, tougher tendons and ligaments exhibit lower elongations, ranging from 10-15% and 10-20%, respectively. Therefore, we believe that not all body parts require elongations exceeding 30%. As mentioned in our paper, FRCIs are intended for applications requiring high strength and toughness, such as fixing artificial bones in robots. The applications and sensing capabilities highlighted in Fig. 4 are primarily targeted at ligaments and similar tissues. Furthermore, even if certain applications demand higher strains, as demonstrated in Fig. 5, the design strategy of FRCIs is versatile. When reinforced with aramid fibers, they exhibit enhanced tensile properties, achieving an elongation of 30%.

Additionally, we are deeply grateful to the Reviewer for suggesting that we carefully consider suitable applications for FRCI with high toughness and tearing resistance. We have further extended its application in self-sensing impact protection. FRCI can effectively resist impact whether or not cracks are present. Furthermore, it has the capability to sense the timing and intensity of impacts, enabling intelligent protection. Therefore, FRCI holds potential in fields requiring high mechanical performance, such as fixing artificial bones and providing impact resistance.

Supplementary Figure 55. Self-sensing impact protection application. (a) Schematic of the impact process. (b) Resistance signals of the impact using steel balls with different weights. (c) Resistance signals of the impact using steel balls with different weights in the presence of cracks. (d) Ionogel underwent rupture after being subjected to impact.

We add the sentence on Page 18 of the revised manuscript.

“Meanwhile, regardless of whether cracks exist in FRCI, it can spontaneously perceive the timing and intensity of external impacts while providing impact protection, thereby enhancing reliability and extending service life (Supplementary Fig. 55a-c). In contrast, traditional ionogels are prone to damage when subjected to excessive impacts (Supplementary Fig. 55d).”

Response to the Reviewer #2

Comment: The authors have well addressed my concerns. In addition, I also evaluated the responses of the authors to the concerns from Reviewer #3. All the issues raised by

Reviewer #3 have been addressed, including the strategy novelty, dissipation energy, the clarity of images, error bars, and the mechanism behind the fiber-reinforced ionogel. Therefore, the current version is recommended for publication.

Reply: We are very grateful to the Reviewer for the encouraging comments on our manuscript.

Response to Reviewers

We are appreciative of the feedback from our peer reviewers. Their constructive comments have significantly contributed to enhancing our manuscript. In response to their concerns, we have expanded our research with additional analyses and discussions. These improvements are thoroughly integrated into the revised manuscript. To facilitate a clear understanding of our revisions, we have organized our responses as follows:

Reviewer Comments and Our Responses: Each comment from the Reviewers (presented in black text) is followed by our corresponding response (in blue text).

Highlighted Text in the Revised Manuscript: Text modifications in the revised manuscript are red in the response letter.

Response to the Reviewer #1

Reply: We are deeply grateful to the Reviewer for the suggestions on our manuscript. We have revised the manuscript according to the Reviewer's suggestion and hope to address all concerns of the Reviewer.

Question 1: For the strain sensor application on human skin, the Young's modulus and bending stiffness should match the modulus of human stratum corneum (~150 kPa), and closely follow the skin under various deformation with robust and reliable attachment between the sensor and skin. On average, biological skin is stretchable to 75% strain, and this allows free movement of the joints, which experience surface strains up to 55% for the knees (Bao et al. Pursuing prosthetic electronic skin. Nature Mater 15, 937 – 950 (2016)). In this work, the Young's modulus is even up to 103-104 MPa in Fig.2i and extremely high bending modulus. Moreover, another point that bothers my understanding is how the FRCI was attached to the knee and elbow (Fig.S49) and artificial bone (Fig.4d) and the authors fail to prove the conformal contact with the epidermis in a manner that does not constrain or alter natural motion or behaviors. Given the extremely high stiffness (and may lack of conformal contact with skin, the authors do not release this information at current stage), I do not think this FRCI is suitable for strain sensor even if they alter the items of ionogel and fabrics in Fig.5.

Answer 1: Thanks for the Reviewer's suggestion. When the material is applied as a

strain sensor on human skin, it indeed requires modulus and compliance similar to those of skin, as you mentioned. However, when the material is used as a ligament analog for fixing joints and performing strain sensing, it needs to possess high modulus, high strength, and high toughness similar to those of ligaments. For instance, Xu *et al.* prepared a ligament-mimicking hydrogel with a modulus of 1.1 GPa through the compositing of aramid nanofibers and hydrogels, which can be used for strain sensing (*Sci. Adv.* **2023**, *9*, eade6973). Dickey *et al.* utilized ionic liquid-toughened glassy gels (with Young's modulus up to 1 GPa), whose resistance also increases with strain, holding promise for application in the field of sensors (*Nature* **2024**, *631*, 313-318). Zhou *et al.* developed hydrogel fibers by mimicking spider silk (with moduli ranging from 1 to 5 GPa), which can serve as strain sensors and have potential applications in artificial tendons (*Adv. Mater.* **2023**, *35*, 2300876). Furthermore, Liang *et al.* created an ultra-sensitive strain sensor using MXene, which also exhibits a high modulus (80 GPa) (*Nat. Commun.* **2024**, *15*, 5354). Therefore, when utilizing materials as strain sensors, the modulus needs to match the application scenario. Low modulus is required when used on skin, whereas high modulus is necessary for biomimetic ligaments. Additionally, when employed as biomimetic ligaments, the fixation method of the material is similar to that reported in the literature, where screws are used to fix it to artificial bones (*Nat. Commun.* **2022**, *13*, 2279). For applications in knees and elbows, adhesive tape is primarily utilized for fixation (*Sci. Adv.* **2023**, *9*, eade6973). Moreover, we emphasize that this high-stiffness material can be applied in the biomimetic ligaments of robots, while low-modulus epidermal sensors require more consideration for conformal contact with skin.

Furthermore, we are deeply grateful to the Reviewer for bringing to our attention the limitation of high-stiffness materials in strain sensor applications. Therefore, we have conducted a detailed discussion on this limitation in the revised main text.

We add the discussion on Page 18 of the revised manuscript.

“Another point worth discussing is that current strain sensors primarily focus on low-modulus flexible materials to enable mechanical matching and conformal contact with the epidermis during use. Therefore, the application of high-stiffness FRCI in epidermal strain sensors may be limited due to its lower elongation compared to biological skin,

coupled with strengths and moduli that are much higher than skin. For FRCI, it may be more suitable for strain sensing in scenarios requiring high strength and high modulus, such as in robotic biomimetic ligaments.”

Question 2: In the author reply, ‘Therefore, this was the main point we intended to convey in our previous reply, namely that we pioneered the toughening of ionogels using fabrics, achieving significantly stronger performance compared to hydrogels, and providing new insights into the design of toughened ionogel materials.’ I think the authors overestimate the potential of their ideal of fabric reinforced ionogels and should tone down this claim. In fact, there are other similar studies published. For example, Huang et al reported a mechanically interlocked ionic gel–elastomer (thermoplastic polyurethanes) hybrid material as soft strain gauge a tiny detection limit of 0.05% strain, ultrafast time resolution of 0.495 ms, and high linearity (<https://advanced.onlinelibrary.wiley.com/doi/10.1002/advs.202301116>). Similarly, Zhang et al applied a direct-ink-write 3D printing process to produce a self-healing and rigid skeleton, and then an ionogel acting soft matrix was injected into the 3D skeleton, and they also specifically examined the resistance to crack growth (<https://advanced.onlinelibrary.wiley.com/doi/10.1002/adma.202405776>). Alternatively, Zhang et al reported a thermoplastic polyurethane nanonet-supported ionogel sensor (<https://advanced.onlinelibrary.wiley.com/doi/10.1002/adfm.202415694>). Also, Jiang et al reported a tough and fatigue-resistant ionic elastomer through the interlocking of a thermoplastic polyurethane (TPU) fibrous scaffold and an ionic supramolecular biopolymer matrix (<https://onlinelibrary.wiley.com/doi/10.1002/anie.202411418>). In this context, as it appears to me that the major unique selling point (reinforcement fabric + soft ionogel) of the present manuscript is incremental, and thus the novelty of the research is questionable, plus the strain sensor application scenario is not suitable due to the extremely high stiffness and may also lack of conformal contact with body, which have been noted above. Although some mechanical performance metric shows their advantage in Fig.2, this work does not adequate to make it innovative enough considering the high standbars of Nat Commun.

Answer 2: Thanks for the Reviewer’s suggestion. We have toned down the previous assertions, but we firmly believe that our work represents a significant advancement

compared to the composite gel materials currently reported. In terms of material design, we employed a universal strategy to establish a tight interfacial bonding between the ionogel and fiber fabric through supramolecular interactions, thereby preventing fiber pull-out. This has resulted in a substantial enhancement in the performance of our composite material, such as its strength, modulus, and tearing toughness, far surpassing that of similar materials (Figure 2). Furthermore, regarding the toughening mechanism, we have discovered that it relies on the synergistic effect between the fiber fabric and ionogel. The supramolecular network within the ionogel not only dissipates energy but also enables the full release of elastic energy within the fibers due to the strong interfacial bonding with the fiber fabric. In terms of applications, we have not only discussed and analyzed the limitations of high-stiffness materials in strain sensors but also envisioned their potential use in scenarios requiring high strength and toughness. Additionally, we have explored their application as impact-resistant materials with sensing capabilities. The above is a compilation and summary of the innovations in our manuscript. We once again express our deepest gratitude to the Reviewer for taking the valuable time to examine our work and provide invaluable suggestions.

Question 3: A minor confusion is they relate the high GF to the significant decrease ionic conductivity over a wide temperature range (page 17), whereas according to the results in Fig.4a, the conductivity increases sharply with increasing of temperature.

Answer 3: Thanks for the Reviewer's question. The ionic conductivity increases with rising temperature. We associate the high GF with a significant decrease in ionic conductivity during deformation (*i.e.*, as strain increases), rather than stating that ionic conductivity decreases with temperature. Studies have shown that for ion-conductive materials, compression of the network during deformation can lead to a reduction in ionic conductivity (*Nat. Commun.* **2022**, *13*, 4411).